# Spot the Fake: Large Multimodal Model-Based Synthetic Image Detection with Artifact Explanation

**Siwei Wen**[1,3]*, **Junyan Ye**[2,1]*, **Peilin Feng**[1], **Hengrui Kang**[4,1],
**Zichen Wen**[4,1], **Yize Chen**[5], **Jiang Wu**[1], **Wenjun Wu**[3], **Conghui He**[1], **Weijia Li**[2,1]†

[1]Shanghai Artificial Intelligence Laboratory, [2]Sun Yat-Sen University,
[3]Beijing Advanced Innovation Center for Future Blockchain and Privacy Computing, Beihang University,
[4]Shanghai Jiao Tong University, [5]The Chinese University of Hong Kong, Shenzhen

## Abstract

With the rapid advancement of Artificial Intelligence Generated Content (AIGC) technologies, synthetic images have become increasingly prevalent in everyday life, posing new challenges for authenticity assessment and detection. Despite the effectiveness of existing methods in evaluating image authenticity and locating forgeries, these approaches often lack human interpretability and do not fully address the growing complexity of synthetic data. To tackle these challenges, we introduce FakeVLM, a specialized large multimodal model designed for both general synthetic image and DeepFake detection tasks. FakeVLM not only excels in distinguishing real from fake images but also provides clear, natural language explanations for image artifacts, enhancing interpretability. Additionally, we present FakeClue, a comprehensive dataset containing over 100,000 images across seven categories, annotated with fine-grained artifact clues in natural language. FakeVLM demonstrates performance comparable to expert models while eliminating the need for additional classifiers, making it a robust solution for synthetic data detection. Extensive evaluations across multiple datasets confirm the superiority of FakeVLM in both authenticity classification and artifact explanation tasks, setting a new benchmark for synthetic image detection. The code, model weights, and dataset can be found here: `https://github.com/opendatalab/FakeVLM`.

## 1 Introduction

As AI-generated content technologies advance, synthetic images are increasingly integrated into our daily lives [1, 2, 3, 4]. Technologies like Diffusion [5, 1, 6] and Flux can generate highly realistic images. However, synthetic image data also poses significant risks, including potential misuse and social disruption [7, 8]. For instance, synthetic fake news [9, 10, 11] and forged facial scams [12, 13] make it increasingly challenging to discern the truth and establish trust in information. In response to these threats, the field of synthetic data detection has garnered widespread attention in recent years. However, traditional synthetic detection methods typically focus on simple authenticity judgments [14, 15, 16, 17, 18, 19], ultimately providing only a forgery probability or classification result. This approach still has significant limitations in terms of human interpretability. As a result, users find it challenging to understand the reasons behind the system's decisions, affecting the decision-making process's transparency and trustworthiness.

The rapid development of large multimodal models has accelerated progress in synthetic data detection [20, 21, 22, 23, 24, 25]. These models, particularly, can provide explanations for authenticity judgments in natural language, thus laying the groundwork for enhancing the interpretability of synthetic

---

*Equal Contribution.
†Corresponding author.

39th Conference on Neural Information Processing Systems (NeurIPS 2025).

detection. For instance, Jia et al. [26] explored ChatGPT's ability to assess synthetic data, highlighting the potential of large models in this domain. LOKI [27] and Fakebench [28] further delved into the capabilities of large models in offering explanations for image detail artifacts. However, these studies primarily focus on pre-trained large models, utilizing strategies such as different prompt wordings to enhance model performance on this task rather than developing a specialized multimodal model for this specific domain. Moreover, existing general large models still show a significant performance gap compared to expert models or human users in detection tasks.

Researchers have further explored and designed multimodal models specifically for synthetic data detection tasks [25, 24, 27, 29, 30]. For instance, works like DD-VQA [23] and FFAA [24] focus primarily on the performance of large models in Deepfake detection tasks, especially in the context of artifact explanation. However, their performance on more general types of synthetic images still requires further investigation. Other works, such as Fakeshield [20] and ForgeryGPT [22], effectively examine the ability of large models to localize forgery artifacts and explain manipulated synthetic data. However, artifacts in forged synthetic images often concentrate in transitional areas, exhibiting more noticeable edge artifacts. In contrast, direct synthetic images are more likely to show structural, distortion, or physical artifacts, highlighting significant differences between the two. Additionally, current large models still lag behind expert models in pure authenticity classification. For example, studies such as X2-DFD [25] and FFAA [24] attempt to integrate traditional expert-based synthetic detection methods to improve classification accuracy without fully unlocking the potential of large models in synthetic detection tasks.

To address the challenges outlined above, we introduce FakeVLM, a large multimodal model specifically designed for fake image detection and artifact explanation. FakeVLM focuses on artifacts generated from synthetic image models rather than forgery artifacts. Moreover, it is not limited to facing Deepfake tasks but extends to more general synthetic data detection. Notably, FakeVLM achieves performance comparable to expert models based on binary classification without requiring additional classifiers or expert models. Additionally, we introduce FakeClue, a dataset containing over 100,000 real and synthetic images, along with corresponding artifact cues for the synthetic images. FakeClue includes images from seven different categories (e.g., animal, human, object, scenery, satellite, document, face manipulation), and leverages category-specific knowledge to annotate image artifacts in natural language using multiple LMMs. Our main contributions are as follows:

- We propose FakeVLM, a multimodal large model designed for both general synthetic and deepfake image detection tasks. It excels at distinguishing real from fake images while also providing excellent interpretability for artifact details in synthetic images.

- We introduce the FakeClue dataset, which includes a rich variety of image categories and fine-grained artifact annotations in natural language.

- Our method has been extensively evaluated on multiple datasets, achieving outstanding performance in both synthetic detection and abnormal artifact explanation tasks.

## 2 Related Work

### 2.1 Synthetic Image Detection

Traditional synthetic detection tasks are typically approached as binary classification tasks [31, 26, 32, 33, 34, 35], treating them as either true or false, based on data-driven methods, with various detectors such as CNNs [36] and transformers [37]. Some research [38, 39, 40] has explored methods like learning anomalies in the frequency domain, reconstruction learning, and adversarial learning. The focus of traditional synthetic detection tasks has mainly been on addressing the generalization issue [15, 16], where the training and testing domains differ, in order to counter the rapidly evolving synthetic models. However, these methods only provide authenticity predictions without offering detailed explanations behind the predictions.

Some works have gone beyond the fundamental true/false classification problem, focusing on interpretable synthetic detection. For example, gradient-based methods are used to visualize highlighted regions of predictions [41, 42, 43]. Alternatively, model designs like DFGNN [44] enhance interpretability by applying interpretable GNNs to deepfake detection tasks. Additionally, research has explored the detection and localization of forgeries by constructing image artifacts or modifying labels [17, 18, 19]. While these methods enhance model interpretability, using natural language to describe the identified reasons remains underexplored.

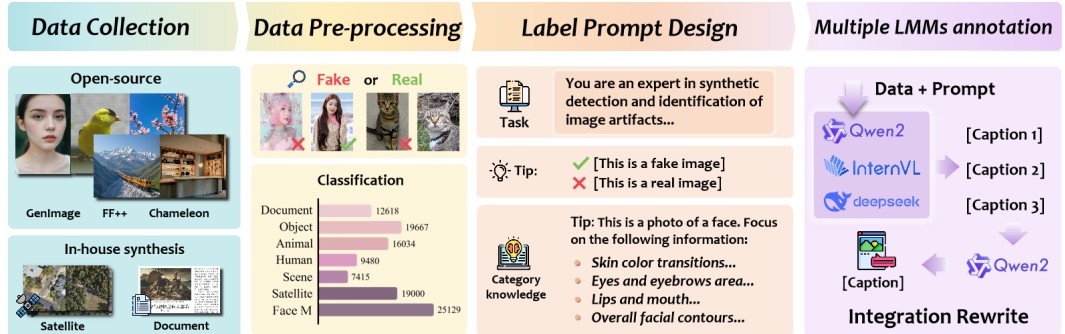

Figure 1: Construction pipeline of FakeClue dataset, including data collection from open source and self-synthesized datasets, pre-processing with categorization, label prompt design based on category knowledge(Face M: Face manipulation), and multiple LMMs annotation with result aggregation.

## 2.2 Synthetic Image Detection via LMMs

Recently, the development of large multimodal models (LMMs) has been rapid [45, 46, 47, 48, 49, 50]. Models such as the closed-source GPT-4o [51] and open-source models like InternVL [52] and Qwen2-VL [53] have demonstrated outstanding performance across various tasks, showcasing impressive capabilities. In the domain of synthetic data detection, several works like [26], Fakebench [28], and LOKI [27] have investigated the potential of LMMs, where these large models not only deliver accurate synthetic detection judgments but also offer natural language explanations for their true/false predictions, enhancing the interpretability of synthetic detection. However, these studies mainly focus on evaluating pretrained large models rather than training expert multimodal models. Furthermore, existing general models still lag behind expert models or humans in detection tasks.

An increasing number of studies have further explored the use of LMMs for synthetic detection, such as DD-VQA [23] and FFAA [24] have explored the performance of large models in the deepfake domain. However, their performance on general synthetic images remains less explored. Fakeshield [20], SIDA [21] and ForgeryGPT [22] investigate the ability of LMMs to detect/explain artifacts in manipulated synthetic data. However, tampering artifacts (mainly transitional), while direct image synthesis artifacts tend to involve structural distortions or other types of image warping, which differ significantly. Furthermore, current LMMs underperform expert models in simple real/false classification tasks [25, 26]. For example, studies like X2-DFD [25] and FFAA [24] have combined traditional expert synthetic detection methods with large models to improve classification accuracy.

# 3 Dataset

## 3.1 Overview

We introduce FakeClue, a multimodal synthetic data detection benchmark for general and DeepFake detection. It includes two tasks: synthetic detection and artifact explanation, requiring models to determine image authenticity and explain its artifacts. FakeClue covers 7 image categories with over 100k samples. Utilizing a multi-LMM labeling strategy with category priors, FakeClue provides image-caption pairs for the image and its natural language artifact explanation. It emphasizes direct synthesis over tampered image artifacts. Training/test sets are randomly split; the test set contains 5,000 diverse image samples. Detailed dataset information is provided in the supplementary materials.

## 3.2 Construction of FakeClue

**Data Collection:** As shown in the data collection phase of Figure 1, FakeClue has two data sources, including open synthetic datasets and our newly synthesized data for specialized types. For open synthetic datasets, we extracted approximately 80K data from GenImage [54], FF++ [55] and Chameleon [56], maintaining a 1:1 ratio of fake to real data. For specialized types of data, such as remote sensing and document images, we generated the data ourselves. We collected remote sensing images from public datasets [57, 58, 59, 60, 61, 62], using GAN and Diffusion-based methods [63, 64], covering urban, suburban, and natural scenes. For document images, we adopted a layout-first, content-rendering approach, generating synthetic images of newspapers, papers, and magazines with real-world data sourced from the M6Doc dataset [65].

**Data Pre-Processing:** At this stage, we first categorize the collected raw image data based on authenticity labels. Since the data from GenImage and Chameleon lack category information, we use a classification model [66] to divide the data into four categories: animal, object, human, and scene. The FF++ dataset corresponds to the face manipulation category, while the newly synthesized satellite and document data also have clear category divisions. The distribution and proportions of these categories are shown in Figure 1. The labels obtained during the data preprocessing stage will serve as the foundation for the subsequent Label Prompt Design.

**Label Prompt Design based on category knowledge:** To overcome the hallucinations and limited synthetic detection capabilities of large models, we inject external knowledge to aid in artifact detection. Pre-processed authenticity labels serve as prior prompt knowledge. For real images, we analyze the plausibility of the image as a photographic result, while for fake images, we focus on detecting artifacts throughout the image. And classification labels are used as category-specific knowledge. This knowledge comprises predefined focal points and common artifact types pertinent to each category, guiding the model's attention to key areas, such as the artifact detection cues for facial images, as shown in Fig. 1. Accordingly, we design *14 distinct types of prompts* to address various authenticity and category labels. In addition, for synthetic images that may have high quality and no visible artifacts (e.g., images from the Chameleon dataset), we use special prompts to prevent the model from forcing interpretation (see Appendix H for detailed prompts for different categories).

**Multiple LMMs Annotation:** To mitigate the bias or hallucination effects of a single multimodal model, we adopt a strategy of annotating and then aggregating results from multiple high-performance open-source large models. For a given image $I_i$ from the set $\{I_i\}_{i=1}^{N}$, we first use three large multimodal models—Qwen2-VL, InternVL, and Deepseek—to generate a set of candidate artifact captions $C_i = \{A_i^1, A_i^2, A_i^3\}$. Each caption $A_i^k \in C_i$ potentially highlights different aspects of image artifacts.

To synthesize a unified annotation $A_i$ from candidate captions $C_i = \{A_i^1, A_i^2, A_i^3\}$, we employ Qwen2-VL for aggregation, denoted as $\mathcal{M}_{\text{agg}}$. This model is prompted with specific instructions $P_{\text{instr}}$ (detailed in the Appendix H) to identify consistent artifacts across $C_i$ and remove redundant information. Thus, the final aggregated annotation $A_i$ for image $I_i$ is obtained as:

$$A_i = \mathcal{M}_{\text{agg}}(C_i, P_{\text{instr}}) \tag{1}$$

Where $A_i^k$ is the $k$-th candidate caption for $I_i$, and $\mathcal{M}_{\text{agg}}$ aggregates $C_i$ using prompt $P_{\text{instr}}$ (enforcing consistency/non-redundancy) into the final annotation $A_i$.

The model extracts common points from multiple model responses, filters out irrelevant observations that appear in only one model (unless they are critical, like glaring artifacts) and organizes them hierarchically by categories (e.g., texture, geometry, lighting), before outputting in a fixed format.

Table 1: Comparison with existing synthetic detection datasets (DF: DeepFake, Gen: General, Syn: Synthesis, Tam: Tampering).

| Dataset | Field | Artifact | Annotator | Category | Number |
|---------|-------|----------|-----------|----------|--------|
| DD-VQA [23] | DF | Syn | Human | | 5k |
| FF-VQA [24] | DF | Syn | GPT | | 95K |
| LOKI [27] | Gen | Syn | Human | ✓ | 3k |
| Fakebench [28] | Gen | Syn | Human | ✓ | 6k |
| MMTD-Set [20] | Gen | Tam | GPT | | 100k |
| FakeClue | DF+Gen | Syn | Multi-LMMs | ✓ | 100k |

## 3.3 Comparisons with Existing Datasets

Table 1 presents a comparison of the synthetic detection performance of FakeClue and existing evaluated LMMs datasets. FakeClue offers broader domain coverage and, unlike DD-VQA, is not confined to DeepFake detection, featuring well-defined categories including specialized types like satellite images and documents. In contrast, LOKI and Fakebench are limited by the number of annotations and can only serve as evaluation sets. Compared to the recent MMTD-Set dataset, FakeClue focuses more on directly synthesized image artifacts rather than tampered artifacts.

# 4 Method

In this section, we first analyze the challenges of large multimodal models in synthetic image detection (Section 4.1). Then, we provide a detailed description of the FakeVLM architecture (Section 4.2) and training strategy (Section 4.3).

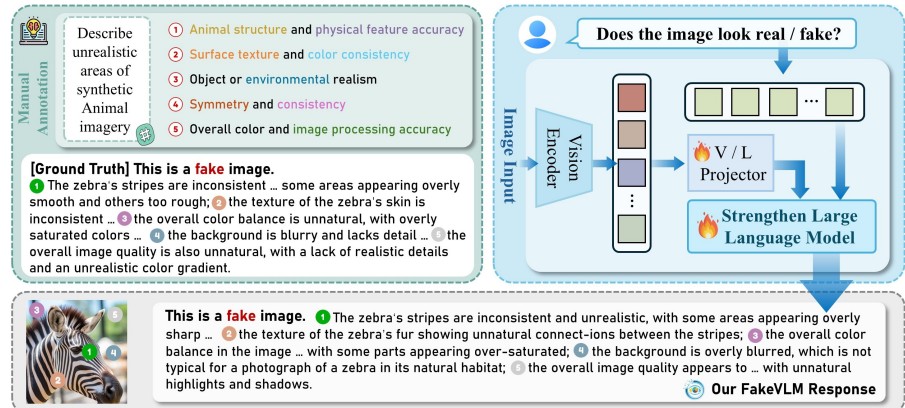

Figure 2: Overview of FakeVLM, our proposed framework for detecting synthetic images and explaining their artifacts. Built upon LLaVA, FakeVLM integrates multiple captioning models to assess key visual aspects.

## 4.1 Re-thinking LMMs' Challenges in Synthetic Image Detection

Recent studies have shown that directly using LMMs for synthetic image detection still face significant challenges [27, 28, 26]. Although large models possess strong text explanation capabilities, when tasked with determining whether an image was AI-generated or identifying forged images from a set, pretrained LMMs often fail to achieve satisfactory performance. This phenomenon highlights the difficulty of relying on LMMs for authenticity judgment, which is closely related to the fact that these models are not inherently designed for synthetic data detection tasks. Nevertheless, through extensive pretraining tasks, multimodal large models have developed strong visual feature extraction abilities and alignment with text. This raises the question: do the internal representations of these large models potentially encode information that can distinguish real images from synthetic ones?

Inspired by Zhang et al. [67], we explored a simple yet effective method on FakeClue: extracting visual features from the last layer of a pre-trained LMM and training a lightweight linear classifier to determine image authenticity. If the representations learned by the model indeed contain discriminative information related to authenticity, even a simple classifier can perform initial detection using these features. As shown in Figure 3, the results confirm that the LMM has potential in distinguishing authenticity. However, in contrast to the conclusions of Zhang et al. [67], framing the task as "Does the image look real/fake?" by using fixed answers like "Real" or "Fake" not only limits the model's ability to provide textual explanations but also results in

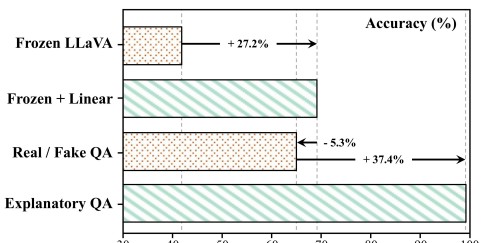

Figure 3: Comparison of synthetic image detection approaches on LOKI and FakeClue datasets: (1) QA with Frozen LMMs (no training), (2) Frozen backbone + linear probe (only linear layer trained), (3) Direct Real/Fake QA tuning, and (4) VQA with artifact explanations tuning.

suboptimal performance. This is likely due to large models' challenges in aligning complex visual content with such binary answers. Building on the dataset we constructed, we found that framing the task as visual question answering, where the model is required to provide not just "Real"/"Fake" answers but also explain the image "artifacts", leads to better alignment between the response text and the image content in Figure 3 (For detailed results, please refer to Appendix B). This approach not only improves the performance of artifact explanation but also significantly enhances the overall performance of synthetic image detection.

## 4.2 Model Architecture

Our approach follows the architecture of LLaVA-v1.5, as illustrated in the framework diagram (see Figure 2 in the top-right corner), which consists of three core components: i) a Global Image Encoder, ii) an MLP Projector, and iii) a Large Language Model (LLM). We detail each component as follows:

**Global Image Encoder**: We employ the pretrained vision backbone of CLIP-ViT(L-14) [68] as our global image encoder. The encoder processes input images with a resolution of 336×336 to preserve synthetic artifact details, resulting in 576 patches per image.

$$V = \text{CLIP-ViT}(I) \in \mathbb{R}^{N \times d_v} \tag{2}$$

where $N = \frac{HW}{P^2}$ denotes the number of patches ($P = 14$), and $d_v = 1024$ the feature dimension.

**Multi-modal Projector**: A two-layer MLP adaptor bridges visual and textual modalities:

$$\begin{aligned} H &= \text{GeLU}(V W_1 + b_1) \\ Z &= H W_2 + b_2 \end{aligned} \tag{3}$$

where $W_1 \in \mathbb{R}^{1024 \times 4096}$, $W_2 \in \mathbb{R}^{4096 \times 4096}$ are learnable parameters. The projected features $Z \in \mathbb{R}^{N \times 4096}$ combine with text embeddings of the task prompt $P$ through concatenation.

**Large Language Model**: We utilize Vicuna-v1.5-7B, a 7B language model, as our base LLM. Vicuna-v1.5 is renowned for its strong instruction-following capabilities and robust performance across diverse tasks. To further enhance its reasoning abilities on synthetic data, we perform full-parameter fine-tuning, optimizing the following objective:

$$\mathcal{L}(\theta) = -\sum_{t=1}^{T_i} \log p_\theta \left( a_{i,t} \mid a_{i,<t}, [Z; E(P)] \right) \tag{4}$$

where $E(\cdot)$ denotes text embeddings. We update all parameters $\theta$ of the LLM during training, enabling full adaptation to synthetic data reasoning while preserving original instruction-following capabilities via full-parameter optimization

By integrating these components, our pipeline leverages the strengths of multimodal models and fine-tunes LLaVA to achieve robust performance in detecting and explaining synthetic data.

## 4.3 Training strategy

As described in the construction process of FakeClue, we leverage category knowledge and utilize multiple LMMs to annotate image artifacts in natural language. As shown on the left side of Fig. 2, we obtain high-quality artifact descriptions as target outputs for the model. Then, we use the QA pairs obtained after the multi-model annotation and summarization steps as our training data. Each data sample consists of: (1) an image $I$; (2) a standardized prompt $P$: "Does the image look real/fake?"; (3) the aggregated answer $A$ from the multi-model annotations.

Our model, initialized from LLaVA-1.5 7B weights [45], underwent full-parameter fine-tuning on our constructed QA dataset. The training is conducted for two epochs on eight NVIDIA A100 GPUs with a batch size of 32 per GPU using a 2e-5 learning rate with 3% linear warmup and cosine decay. This full fine-tuning adapted the model to synthetic data detection/explanation nuances while preserving its general instruction-following capabilities.

# 5 Experiment

In this section, we introduce three additional datasets used in the experiments, alongside FakeClue, and describe our experimental setup. We then present FakeVLM's performance on general synthetic and DeepFake detection tasks, as well as its ability to explain image artifacts. Finally, we conduct ablation studies and further exploratory experiments to assess the model's performance.

## 5.1 Other Benchmarks

**LOKI [27]** is a recently proposed benchmark for evaluating multimodal large models in general synthetic detection tasks. Beyond just distinguishing real from fake, LOKI also includes **human manually annotated** fine-grained image artifacts, enabling a thorough exploration of the model's ability to explain image artifacts. This inclusion allows us to verify, to some extent, whether our model's perception of artifacts aligns with human cognition.

**FF++ [55]** is a widely used benchmark dataset for facial forgery detection, containing face images and videos generated by different types of forgery techniques. The dataset includes forged data created using four common forgery methods: DeepFakes, Face2Face, FaceSwap, and NeuralTextures. We used the commonly employed C23 versions.

**DD-VQA [23]** is a new face-domain artifact explanation dataset leveraging human common-sense perception for authenticity assessment. It features artifacts like blurred hairlines and unnatural skin shadows. Built on FF++ data, DD-VQA employs **manual artifact annotations** in a VQA format, requiring models to answer common-sense questions about artifacts.

## 5.2 Experimental setup

**Task Settings.** LOKI serves as the evaluation set, while training is conducted on the FakeClue dataset, with testing performed on LOKI. For FF++ and DD-VQA, we use their default training-test splits for evaluation. Evaluation metrics cover two tasks: detection and artifact explanation. Classification accuracy is represented by Acc, Auc, and F1 scores, while artifact explanation accuracy is measured using CSS and ROUGE_L.

**Compared Baselines.** For tasks requiring both synthetic detection and artifact explanation (e.g., FakeClue, DD-VQA, LOKI), we compared various general-purpose LMMs, including closed-source models like GPT-4 and open-source ones such as Qwen2-VL, LLaVA, InternVL2, and Deepseek-VL2. We also included the Common-DF method from the DD-VQA dataset. For pure synthetic data detection, we further compared with recent SOTA expert methods.

## 5.3 Universal synthetic detection

Table 2 compares FakeVLM with leading general-purpose LMMs and expert models, demonstrating its superior performance in both synthetic detection and artifact explanation. Specifically, compared to the current powerful open-source model Qwen2-VL-72B and leading expert model NPR or AIDE, which is also trained on FakeClue, FakeVLM achieves an average improvement of 7.7% in Acc and 3.6% in F1 on both FakeClue and LOKI. Additionally, LOKI includes an extra evaluation metric for human performance, with an Acc of 80.1%, whereas FakeVLM achieves an Acc of 84.3%, surpassing

Table 2: The experimental results on the FakeClue and LOKI datasets include both Detection and Artifact Explanation performance. ∗ denotes methods trained on FakeClue.

| Method | FakeClue (Ours) | | | | LOKI Evaluations (ICLR 2025) | | | |
|---|---|---|---|---|---|---|---|---|
| | Acc ↑ | F1 ↑ | ROUGE_L ↑ | CSS ↑ | Acc ↑ | F1 ↑ | ROUGE_L ↑ | CSS ↑ |
| Deepseek-VL2-small [69] | 40.4 | 54.2 | 17.1 | 50.4 | 25.3 | 38.7 | 16.4 | 39.1 |
| Deepseek-VL2 [69] | 47.5 | 54.1 | 17.2 | 50.5 | 43.1 | 39.2 | 16.9 | 38.8 |
| InternVL2-8B [52] | 50.6 | 49.0 | 18.0 | 58.1 | 52.6 | 34.0 | 17.9 | 47.2 |
| InternVL2-40B [52] | 50.7 | 46.3 | 17.6 | 55.2 | 50.7 | 37.6 | 18.4 | 47.3 |
| Qwen2-VL-7B [53] | 45.7 | 59.2 | 26.6 | 56.5 | 57.1 | 35.0 | 18.2 | 38.4 |
| Qwen2-VL-72B [53] | 57.8 | 56.5 | 17.5 | 54.4 | 55.4 | 40.9 | 17.3 | 43.2 |
| GPT-4o (2024-08-06) [51] | 47.4 | 42.0 | 13.4 | 40.7 | 63.4 | 57.2 | 14.7 | 35.4 |
| CNNSpot [70] | 43.1 | 9.8 | - | - | 43.1 | 11.4 | - | - |
| FreqNet [71] | 48.7 | 39.3 | - | - | 58.9 | 50.6 | - | - |
| Fatformer [72] | 54.5 | 45.1 | - | - | 58.8 | 48.4 | - | - |
| UnivFD [15] | 63.1 | 46.8 | - | - | 49.0 | 35.8 | - | - |
| AIDE∗ [56] | 85.9 | 94.5 | - | - | 65.6 | 80.2 | - | - |
| NPR∗ [73] | 90.2 | 91.6 | - | - | 77.4 | 80.0 | - | - |
| FakeVLM | **98.6** | **98.1** | **58.0** | **87.7** | **84.3** | **83.7** | **20.1** | **58.2** |

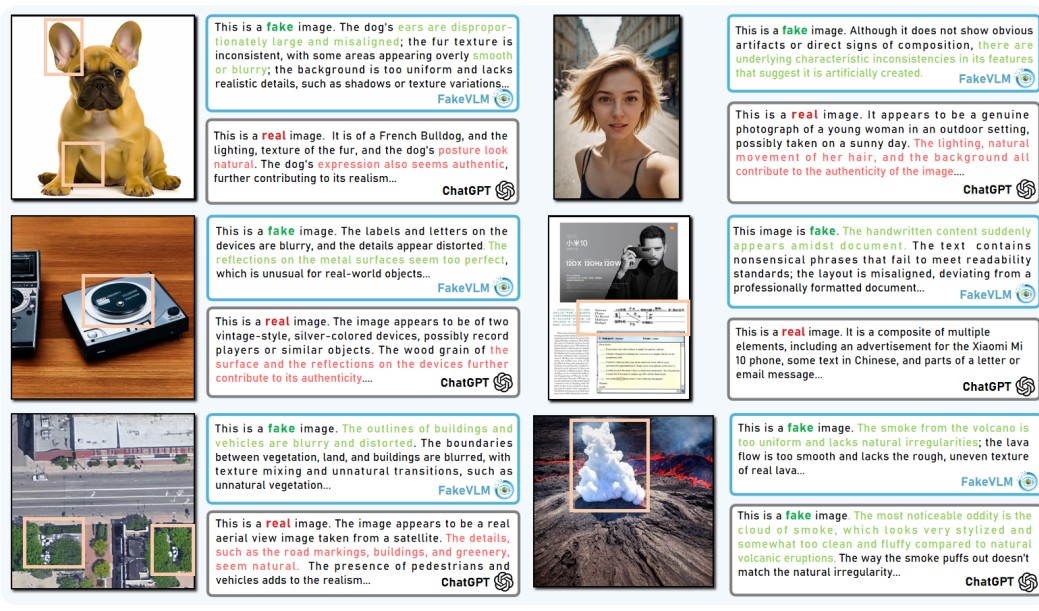

Figure 4: Synthetic image detection cases, covering animals, people, objects, documents, and remote sensing (red denotes incorrect, green denotes correct detection). FakeVLM outperforms GPT in precision, comprehensiveness, and relevance, demonstrating superior detection and interpretation.

human performance. This is likely attributed to FakeVLM's ability to capture deep, image-level features that are imperceptible to the human eye for accurate authenticity judgment. Moreover, FakeVLM's strong ROUGE_L and CSS scores on the human-annotated LOKI dataset highlight alignment with human artifact perception, a noteworthy achievement given its training on entirely LMM-generated data, which is largely thanks to our meticulously designed data annotation pipeline.

Figure 4 presents FakeVLM's qualitative evaluation. FakeVLM identifies synthesis issues (e.g., visual artifacts, texture distortions, structural anomalies) with detailed natural language explanations. Unlike probability-threshold methods, FakeVLM's intuitive, interpretable descriptions, enhance detection transparency, enabling confident and reliable synthetic content assessment.

We also present the generalization experiment results of FakeVLM on DMimage [74] in Table 3, following Huang et al. [21]. The experimental results show that the performance gap between FakeVLM and other expert models is not significant, with FakeVLM even outperforming some of them. FakeVLM does not rely on additional classifiers or expert models, yet it achieves performance comparable to or even exceeding that of expert classifiers while retaining its language capability for artifact explanation. This demonstrates the potential of large models in synthetic detection.

Table 3: Comparison with other detection methods on the DMimage [74] dataset, using the original weights for each method.

| Method | Real | | Fake | | Overall | |
|---|---|---|---|---|---|---|
| | Acc | F1 | Acc | F1 | Acc | F1 |
| CNNSpot [70] | 87.8 | 88.4 | 28.4 | 44.2 | 40.6 | 43.3 |
| Gram-Net [75] | 62.8 | 54.1 | 78.8 | 88.1 | 67.4 | 79.4 |
| Fusing [8] | 87.7 | 86.1 | 15.5 | 27.2 | 40.4 | 36.5 |
| LNP [76] | 63.1 | 67.4 | 56.9 | 72.5 | 58.2 | 68.3 |
| UnivFD [15] | 89.4 | 88.3 | 44.9 | 61.2 | 53.9 | 60.7 |
| AntifakePrompt [77] | 91.3 | 92.5 | 89.3 | 91.2 | 90.6 | 91.2 |
| SIDA [21] | 92.9 | 93.1 | 90.7 | 91.0 | 91.8 | 92.4 |
| FakeVLM | **98.2** | **99.1** | **89.7** | **94.6** | **94.0** | **94.3** |

## 5.4 DeepFake detection

We evaluate FakeVLM's performance on DeepFake detection, with results on DD-VQA presented in Table 4. FakeVLM surpasses both general-purpose multimodal models and the specialized vision-language model, Common-DF, with improvements of 5.7% in Acc, 3% in F1, and 9.5 % in ROUGE_L.

Table 4: The experimental results were evaluated on the DD-VQA datasets. Common-DF-T and Common-DF-I represent text or image contrastive losses, respectively, while Common-DF-TI denotes both text and image contrastive losses.

| Method | DD-VQA (ECCV 2024) | | | |
| --- | --- | --- | --- | --- |
| | Acc ↑ | F1 ↑ | ROUGE_L ↑ | CSS ↑ |
| InternVL2-8B [52] | 56.9 | 53.1 | 14.3 | 51.1 |
| InternVL2-40B [52] | 52.5 | 57.7 | 22.2 | 54.5 |
| Qwen2-VL-7B [53] | 45.7 | 58.9 | 26.6 | 56.5 |
| Qwen2-VL-72B [53] | 59.5 | 57.9 | 20.5 | 56.6 |
| GPT-4o [51] | 53.2 | 31.7 | 13.0 | 42.7 |
| Common-DF-T [78] | 83.7 | 87.6 | 57.7 | - |
| Common-DF-I [78] | 84.9 | 88.4 | 58.8 | - |
| Common-DF-TI [78] | 87.5 | 90.1 | 60.9 | - |
| FakeVLM | **93.2** | **93.1** | **70.4** | **86.6** |

Table 5 shows FakeVLM's evaluation on the FF++ DeepFake detection dataset, including sub-categories like DeepFakes, Face2Face, FaceSwap, and NeuralTextures. The experimental results demonstrate that FakeVLM continues to exhibit strong performance in these tasks, comparable to or even surpassing leading deepfake expert models. It not only leads in overall detection but also exhibits balanced performance across categories, avoiding overfitting.

Table 5: Performance evaluation of FakeVLM on the FF++ DeepFake detection dataset. The results highlight FakeVLM's robust detection performance, on par with or outperforming top expert models.

| Method | FF++ (ICCV 2019) - AUC(%) | | | | |
| --- | --- | --- | --- | --- | --- |
| | FF-DF | FF-F2F | FF-FS | FF-NT | Average |
| FWA [79] | 92.1 | 90.0 | 88.4 | 81.2 | 87.7 |
| Face X-ray [80] | 97.9 | 98.7 | 98.7 | 92.9 | 95.9 |
| SRM [81] | 97.3 | 97.0 | 97.4 | 93.0 | 95.8 |
| CDFA [82] | 99.9 | 86.9 | 93.3 | 80.7 | 90.2 |
| FakeVLM | 97.2 | 96.0 | 96.8 | 95.0 | 96.3 |

## 5.5 Ablation Study and More Exploration

**Impact of explanatory text.** To validate the effectiveness of our explanatory VQA text paradigm, we conduct ablation experiments comparing two variants under the same training settings: (1) LLaVA + Linear Head: Full parameter fine-tuning with a linear classification head trained for binary prediction; (2) LLaVA + Explanatory Text: Training LLaVA to generate explanatory answers instead of fixed labels. Both variants trained on FakeClue, tested on LOKI; all parameters trainable. As shown in Table 6, the explanatory text paradigm demonstrates advantages in out-of-distribution (OOD) generalization on the LOKI benchmark.

Table 6: Ablation study comparing linear classification and explanatory text paradigms.

| Method | LOKI Evaluations (ICLR 2025) | | | |
| --- | --- | --- | --- | --- |
| | Acc ↑ | F1 ↑ | ROUGE_L ↑ | CSS ↑ |
| LLaVA + Linear Head | 81.6 | 77.8 | - | - |
| LLaVA + Explanatory Text | 84.3 | 80.1 | 60.9 | 58.2 |

**Performance on Real Images.** In practical applications, authentic images dominate, necessitating models that can avoid misidentifying artifacts and incorrectly filtering genuine images. Fig 5 illustrates FakeVLM's real image performance, showing it integrates features like object structure, lighting, color, and fine textures for authenticity assessment. This holistic approach enhances the model's reliability and robustness in distinguishing between synthetic and real images. Additional experiments, including robustness studies on image perturbations, are provided in Appendix C.

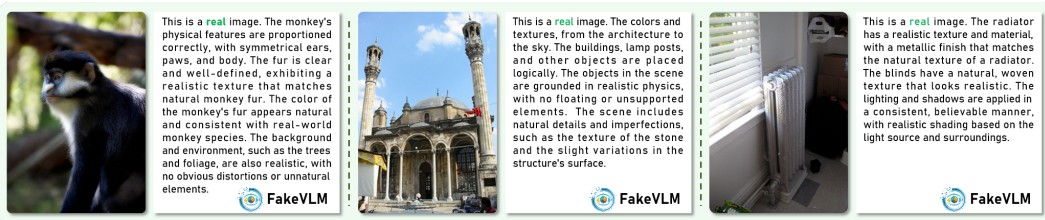

Figure 5: Performance of FakeVLM on real images.

## 6 Conclusion

The rapid growth of AI-generated images has posed challenges to the authenticity of information, driving the demand for reliable and transparent detection methods. As image synthesis detection techniques have advanced alongside large multimodal models (LMMs), approaches have shifted from non-LMMs to methods based on LMMs. Our proposed FakeVLM is a large model that integrates both synthetic image detection and artifact explanation. Through an effective training strategy, FakeVLM leverages the potential of large models for synthetic detection without relying on expert classifiers. It performs well in both synthetic detection and artifact explanation tasks, offering new insights and directions for future research in synthetic image detection.

## Acknowledgments

This work was partially supported by the National Natural Science Foundation of China (Grant No. 62571560, 62476016 and 62441617), Shanghai Artificial Intelligence Laboratory and Beijing Advanced Innovation Center for Future Blockchain and Privacy Computing.

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

# Appendices

## A    Dataset visualization examples

Figure 6 shows some example images from the FakeClue dataset, which includes seven major categories. Our dataset also contains a rich variety of synthetic images with different qualities and resolutions, some of which exhibit noticeable artifacts, while others are difficult to detect with the naked eye. The typical artifact characteristics vary across different categories of images, which is why we use different prompts tailored to each category in order to enhance the model's ability to capture these artifacts.

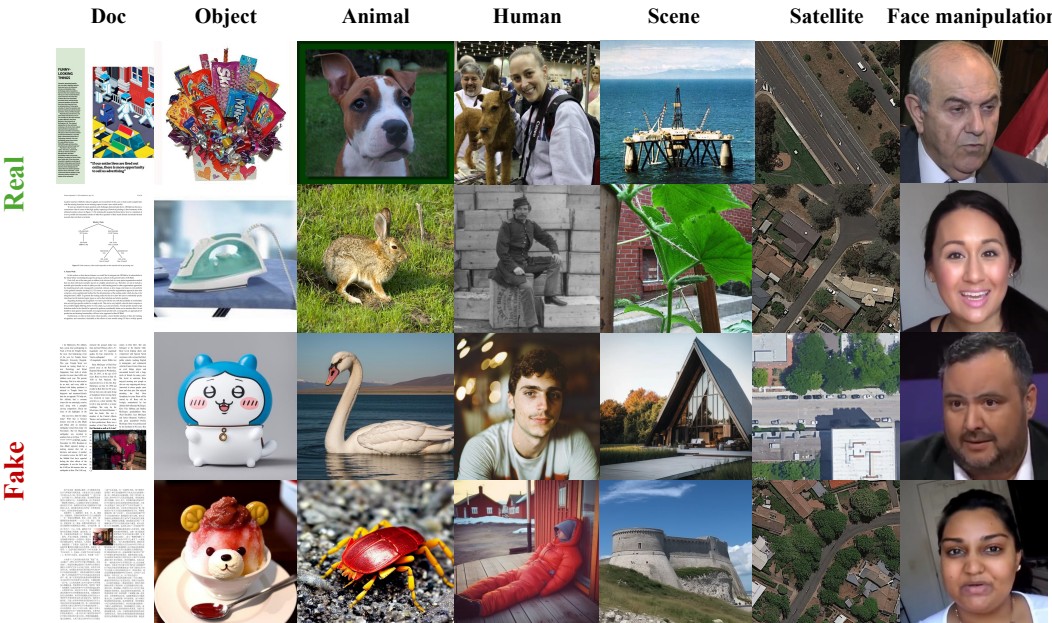

Figure 6: Real and fake image examples from the FakeClue dataset, categorized by Document, Object, Animal, Human, Scene, Satellite, and Face Manipulation.

## B    Different training strategies

We explore different training strategies on FakeClue and evaluate them on both the FakeClue test set and the additional LOKI test set. As shown in Figure 7, a simple linear layer can partially activate the large model's capability in synthetic detection. Compared to the direct Real/Fake QA approach, the VQA format, which includes artifact explanations alongside authenticity judgments, achieves the best performance. This improvement is likely due to better alignment between visual image content and textual explanations.

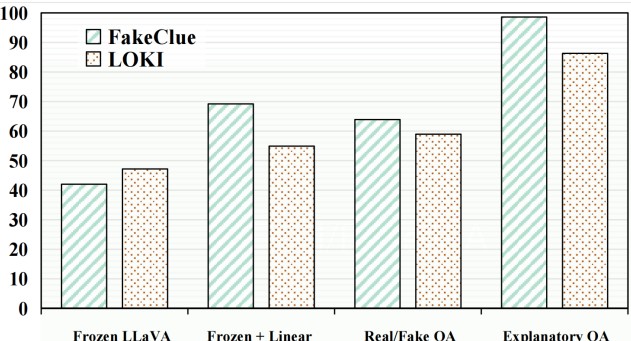

Figure 7: Performance of different training strategies.

## C   Robustness Study

We further evaluated the robustness of FakeVLM to common image distortions, such as JPEG compression, resizing, and Gaussian noise. Table 7 presents the performance of our model on FakeClue under eight distortion scenarios: JPEG compression (quality levels 70 and 80), resizing (scaling factors of 0.5 and 0.75), Gaussian noise (variances of 5 and 10), as well as additional transformations such as flipping, rotating, sharpening, and adjusting contrast or blur levels. Although the model was not explicitly trained on distorted data, FakeVLM demonstrated resilience to these low-level distortions, highlighting its robustness and practical applicability.

Table 7: Performance of FakeVLM under different perturbations on FakeClue.

| Method | Detection | | Explanation | |
|---|---|---|---|---|
| | Acc ↑ | F1 ↑ | ROUGE_L ↑ | CSS ↑ |
| JPEG 70 | 91.2 | 88.7 | 55.7 | 85.3 |
| JPEG 80 | 91.0 | 88.5 | 56.3 | 85.9 |
| Resize 0.5 | 96.4 | 94.9 | 57.0 | 87.0 |
| Resize 0.75 | 98.1 | 97.4 | 57.7 | 87.5 |
| Gaussian 10 | 92.1 | 89.8 | 56.1 | 86.4 |
| Gaussian 5 | 94.5 | 92.5 | 56.5 | 86.6 |
| Flip orizontal | 98.4 | 98.3 | 54.0 | 83.3 |
| Rotate15 | 90.9 | 89.6 | 44.0 | 76.6 |
| Sharpen1.5 | 97.4 | 97.2 | 54.3 | 83.6 |
| Contrast0.7 | 96.4 | 96.1 | 53.0 | 82.8 |
| Contrast1.3 | 98.2 | 98.0 | 54.0 | 83.3 |
| Blur3 | 89.5 | 87.8 | 50.3 | 81.2 |
| Origin image | 98.6 | 98.1 | 58.0 | 87.7 |

## D   Category generalization test

In the LOKI evaluation set, different category divisions allow us to assess the model's performance across various categories. Table 8 presents the performance of FakeVLM and GPT-4o on different image categories. The results indicate that GPT-4o exhibits a noticeable category bias, whereas FakeVLM, trained on diverse data, achieves a more balanced performance across all categories. Additionally, its relatively lower performance on human portraits may be attributed to the rapid advancements in generative models within this domain.

## E   Performance Summary

We visualize the performance of FakeVLM and several leading LMMs across three datasets—DD-VQA [23], FakeClue, and LOKI [27]—using a radar chart, as shown in Figure 8. The results indicate that FakeVLM demonstrates a clear advantage over existing general-purpose multimodal models in both synthetic detection and artifact explanation tasks.

## F   Limitations and Future Works

Despite the strong performance of FakeVLM and the scalability of the FakeClue dataset, certain limitations warrant attention. First, as FakeClue's annotations are primarily derived from multi-LMM aggregation, potential biases intrinsic to the annotating models and insufficient capture of fine-grained artifacts may persist. Future iterations should incorporate heterogeneous annotation strategies, including human expert validation, to enhance dataset robustness. Second, FakeVLM exhibits diminished sensitivity to high-fidelity synthetic images with imperceptible artifacts. Detecting such subtle inconsistencies demands more advanced methodologies capable of analyzing latent statistical irregularities beyond conventional artifact cues.

Table 8: **Judgment** questions results of different models on the LOKI **Image** modality. [*] denotes the closed-source models.

| | Overall | Scene | Animal | Person | Object | Medicine | Doc | Satellite |
|---|---|---|---|---|---|---|---|---|
| Expert (AIDE) | 63.1 | - | 89.9 | 62.5 | 96.5 | 53.4 | 49.7 | 39.3 |
| MiniCPM-V-2.6 | 44.8 | 52.0 | 34.4 | 53.1 | 31.5 | 53.8 | 51.5 | 38.3 |
| Phi-3.5-Vision | 52.5 | 50.8 | 41.7 | 71.5 | 34.1 | 57.3 | 54.3 | 60.5 |
| LLaVA-OneVision-7B | 49.8 | 59.2 | 41.9 | 58.1 | 37.3 | 52.3 | 53.0 | 50.1 |
| InternLM-XComposer2.5 | 46.4 | 52.7 | 40.0 | 56.7 | 32.5 | 56.1 | 49.8 | 38.2 |
| mPLUG-Owl3-7B | 45.9 | 52.1 | 37.3 | 52.9 | 31.4 | 55.3 | 53.8 | 38.1 |
| Qwen2-VL-7B | 47.8 | 54.7 | 38.9 | 57.9 | 30.3 | 56.0 | 59.6 | 36.9 |
| LongVA-7B | 46.2 | 57.6 | 37.4 | 52.5 | 34.1 | 54.4 | 49.8 | 39.7 |
| Mantis-8B | 54.6 | 54.9 | 52.2 | 54.8 | 53.5 | 53.1 | 51.9 | 63.3 |
| Idefics2-8B | 45.0 | 51.8 | 35.3 | 52.3 | 29.2 | 52.3 | 53.9 | 40.6 |
| InternVL2-8B | 49.7 | 58.8 | 39.4 | 54.4 | 37.8 | 53.9 | 60.2 | 44.2 |
| Llama-3-LongVILA-8B | 49.8 | 49.8 | 50.5 | 50.6 | 47.2 | 50.0 | 49.9 | 50.0 |
| VILA1.5-13B | 49.3 | 52.0 | 38.6 | 54.2 | 31.0 | 50.1 | 56.6 | 62.4 |
| InternVL2-26B | 44.3 | 51.6 | 35.4 | 50.8 | 28.2 | 51.3 | 54.4 | 37.6 |
| VILA1.5-40B | 48.8 | 53.7 | 39.3 | 50.0 | 33.4 | 52.5 | 59.9 | 50.6 |
| InternVL2-40B | 49.6 | 55.7 | 37.3 | 59.2 | 34.8 | 55.5 | 64.8 | 40.8 |
| Qwen2-VL-72B | 53.2 | 55.9 | 43.4 | 66.9 | 38.0 | 55.9 | 73.7 | 38.2 |
| LLaVA-OneVision-72b | 46.3 | 54.7 | 31.6 | 53.1 | 27.8 | 52.1 | 67.9 | 36.6 |
| Claude-3.5-Sonnet[*] | 53.6 | 51.6 | 51.6 | 55.2 | 51.4 | 51.9 | 59.1 | 50.9 |
| Gemini-1.5-Pro[*] | 43.5 | 53.7 | 35.7 | 51.5 | 30.3 | 50.0 | 47.2 | 38.1 |
| GPT-4o[*] | 63.4 | 70.1 | 69.7 | 84.4 | 70.3 | 54.3 | 60.1 | 45.0 |
| FakeVLM | 84.3 | 98.9 | 89.7 | 73.8 | 94.1 | 69.3 | 85.0 | 97.2 |

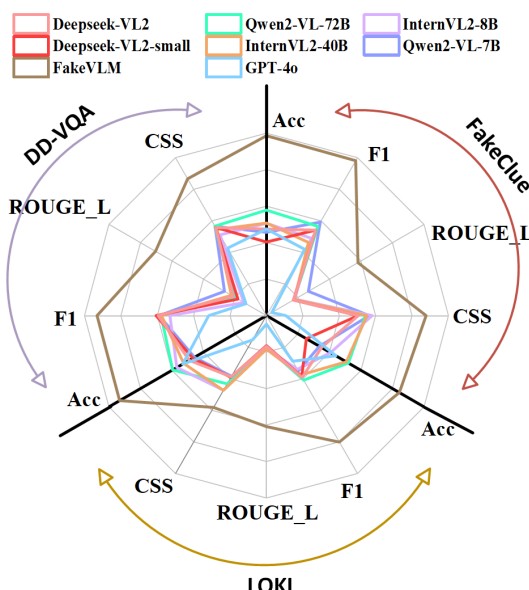

Figure 8: The performances of the 7 leading LMMs on DD-VQA [23], FakeClue and LOKI [27].

## G   Broader Impacts

This study investigates the use of Large Multimodal Models (LMMs) for synthetic image detection and artifact interpretation, offering both significant societal benefits and potential ethical challenges. On the positive side, tools such as FakeVLM enhance the identification of AI-generated content and provide interpretable artifact explanations, thereby promoting media authenticity, digital trust, and public awareness. These capabilities are critical in countering misinformation, forged visual

content, and deceptive practices, with added value in user education through transparency of detection rationale.

However, the approach also raises important concerns. As generative techniques evolve in parallel with detection methods, a dynamic adversarial escalation may ensue, potentially resulting in more evasive synthetic content. Moreover, false positives risk unjust censorship or reputational harm, while false negatives permit malicious media to persist. Overreliance on automated systems may also weaken critical user discernment. Additionally, the computational intensity of LMMs could restrict their adoption to resource-rich institutions, exacerbating inequality in digital verification capabilities.

Thus, while the proposed framework represents a step toward securing online visual ecosystems, responsible deployment necessitates further research, ethical foresight, and inclusive public education to mitigate associated risks.

## H   Label Prompt

At the end of the supplementary materials, we provide specific examples of the Label Prompts mentioned in the main text. These examples cover different categories, including Animal, Scene, Object, Human, Satellite, and Document, with variations for both real and fake labels. Additionally, we detail the specific prompt designed for the aggregation of outputs from the three initial multimodal models.

## Model output merging

You are an expert in AI-generated content analysis. Merge the following three model responses into one unified answer. All responses agree on the image's authenticity (real/fake). Prioritize explanations mentioned by at least two models and omit points unique to a single model.
Follow these steps:

- **Extract Common Ground:** Identify overlapping details (e.g., shadows, outlines, object alignment) across all three responses.

- **Filter Minority Claims:** Discard observations mentioned by only one model unless they are critical (e.g., glaring artifacts).

- **Structure Hierarchically:** Group explanations by category (e.g., lighting, geometry, textures) for clarity.

- **Maintain Original Format:** Begin with "This is a [real/fake] image." followed by a concise, semicolon-separated list of consolidated evidence.

- **Avoid Redundancy:** Rephrase overlapping points to eliminate repetition while preserving technical accuracy.

- **Ensure Logical Consistency:** If any response contains nonsensical, contradictory, or infinite loop reasoning, disregard that portion of the answer.

Output format:
For a real image:
```
"This is a real image.  [Explanation summarizing common realistic features,
such as clear outlines, organized objects, realistic shadows, and aligned
roads.]"
```
For a fake image:
```
"This is a fake image.  [Explanation summarizing common unrealistic features,
such as blurry outlines, disorganized objects, unrealistic shadows, and
misaligned roads.]"
```

## animal

**# Real**

<image>Describes realistic areas of real animal imagery.

Note:

- **Animal structure and physical feature accuracy:** Check if the animal's features like ears, paws, wings, etc., are correctly proportioned and symmetrical. Ensure the animal's body parts are connected in a natural and realistic manner.

- **Surface texture and color consistency:** Examine if the animal's texture is realistic, with clear, well-defined fur, feathers, or skin. The texture should match natural details, and the colors should be accurate and natural, without unnatural blending or over-saturation.

- **Object or environmental realism:** Check the background and surroundings for natural elements, ensuring the animal is integrated with its environment in a way that reflects the real world. The connections between objects, like water, plants, or terrain, should be believable.

- **Symmetry and consistency:** Look for symmetry in the animal's body parts, with natural connections between joints, paws, or wings. The overall shape and structure should be consistent without any noticeable discrepancies.

- **Overall color and image processing accuracy:** Ensure the overall color balance in the image is natural and consistent with real-world animal features, avoiding any unrealistic color shifts or over-saturation.

Output format:

Please output your answer in non-segmented text format, not Markdown format, as follows:
`"This is a real image.  [Explanation of the realistic aspects found in the image, such as accurate animal structure, natural texture, realistic environment, etc.]"`

**# Fake**

<image>Describes unrealistic areas of synthetic animal imagery.

Note:

- **Animal structure and physical feature anomalies:** Check if the animal's features like ears, paws, wings, etc., are mismatched or distorted. Ensure the proportions of animal parts (like paws, wings, and tails) are accurate and symmetrical. Look for unnatural connections between body parts.

- **Surface texture and color distortion:** Examine if the animal's texture is inconsistent, such as blurry patches or unnatural blending. Ensure the fur, feathers, or skin texture matches realistic details. Look for overly saturated or unnatural colors in the animal's appearance.

- **Object or environmental anomalies:** Check the background for abnormal noise, holes, or disjointed textures. Ensure objects in the environment (like water, plants, or terrain) are realistically connected to the animal. Look for unnatural reflections or highlights, especially in aquatic settings.

- **Symmetry and consistency issues:** Look for asymmetry or noticeable discrepancies in the animal's parts. Ensure the connections between body parts (like joints, paws, or wings) are clearly defined and consistent.

- **Overall color and image processing distortion:** Ensure the overall color balance in the image looks natural and not over-saturated. Watch for color shifts that don't align with real-world animal features.

Output format:

Please output your answer in non-segmented text format, not Markdown format, as follows: `"This is a fake image.  [Explanation of the issues found in the image, such as animal structure anomalies, texture inconsistencies, unnatural environment, etc.]"`

<Few-shot Examples >

# Real

<image>Describes realistic areas of real animal imagery.

Note:

- **Physical logic accuracy:** Check if all objects in the scene are grounded in realistic physics, with no floating or unsupported elements. Ensure that all objects are placed in logical positions and follow natural laws, such as trees with visible roots or furniture resting on solid surfaces.

- **Structural and environmental consistency:** Look for harmony between the objects and their environment. Ensure that the placement of structures like houses or roads makes sense within the context of the surrounding landscape. For instance, a house should have a stable foundation, and roads should blend naturally with the terrain.

- **Consistent lighting and shadows:** Observe if the lighting and shadows align logically with one another, with clear and natural light sources. Ensure that shadows cast by objects are consistent in direction and intensity, and that lighting across different parts of the image (such as the foreground and background) is coherent.

- **Natural color and texture fidelity:** Check if the colors are realistic and not overly saturated. Ensure that materials like grass, water, or rocks have natural, believable textures. The terrain and other elements should reflect real-world characteristics without unnatural distortions.

- **Clear edges and smooth transitions:** Ensure that all transitions between objects and environments are realistic and clear, without unnatural blending. For example, a shoreline should transition naturally into water, and terrain features should merge in a way that makes sense within the landscape.

- **Natural details and imperfections:** Look for small, realistic imperfections or variations in the scene, such as asymmetry in plant growth, irregularities in rock formations, or natural inconsistencies in structure shapes. Ensure that the details align with how things appear in the real world, avoiding perfect symmetry or overly manufactured features.

Output format:

Please output your answer in non-segmented text format, not Markdown format, as follows: "This is a real image. [Explanation of the realistic aspects found in the image, such as physical accuracy, consistent lighting, natural textures, etc.]"

# Fake

<image>Describes unrealistic areas of synthetic scene imagery.

Note:

- **Physical logic anomalies:** Check if objects appear to defy natural laws, such as floating chairs or trees with exposed roots. Ensure elements of the scene are grounded in realistic physics, with no floating or unsupported objects. Look for any objects in impossible positions or states that would not naturally occur.

- **Structural and environmental contradictions:** Look for mismatches between the objects and their environments. For instance, check if a house is placed in an odd location, like on a rocky surface with no foundation, or if a smooth road appears in an otherwise rugged landscape. Ensure that the placement of objects makes sense within the context of the environment.

- **Inconsistent lighting and shadows:** Observe if lighting and shadows do not match, such as light sources coming from inconsistent directions, strange reflections, or broken shadows. Check if areas of the image, particularly the upper and lower portions, have lighting that doesn't align with one another.

- **Color and texture distortion:** Check if the colors are overly saturated or unrealistic. For instance, unnatural shades of green for grass or unrealistic wave textures. Pay attention to materials that appear distorted or do not resemble real-world textures, such as block-like rocks or unnatural terrain features.

- **Edge and transition anomalies:** Look for unnatural or blurred boundaries between objects and environments. Check for smooth or illogical transitions, such as a shoreline blending too smoothly into water or an unnatural merge between terrain features. Make sure all transitions between objects and the environment are clear and realistic.

- **Detail violations of reality:** Examine whether details are too perfect or unnatural. For instance, overly symmetrical flower arrangements, distorted structures like leaning towers, or plants that grow in unrealistic patterns. Ensure that all details in the scene align with natural or realistic conditions.

Output format:

Please output your answer in non-segmented text format, not Markdown format, as follows: "This is a fake image. [Explanation of the issues found in the image, such as physical anomalies, structural contradictions, color distortions, etc.]"

object

# Real

<image>Describes realistic areas of real object imagery.

Note:

- **Material and texture authenticity:** Check if the surface texture of objects reflects real-world materials accurately, such as leaves that look like real leaves or a keyboard with clearly defined key textures. Ensure the textures appear clear and natural, corresponding to the material's properties.

- **Consistent shapes and structures:** Ensure that objects have natural and realistic shapes, with no distortion or unnatural features. For instance, a keyboard should have a well-defined spacebar, and a boat should not be stacked on top of another in a way that defies real-world physics.

- **Accurate color and lighting:** Observe whether the colors of objects are natural and not overly saturated, and ensure that the lighting and shadows are applied in a consistent, believable manner. Objects should have realistic shading based on light sources and their surroundings.

- **Natural color and texture fidelity:** Check if the colors are realistic and not overly saturated. Ensure that materials like grass, water, or rocks have natural, believable textures. The terrain and other elements should reflect real-world characteristics without unnatural distortions.

- **Sharp and clear details:** Ensure that all details in the image, such as text on a device or the edges of objects, are sharp, well-defined, and easy to interpret. There should be no blurriness or loss of clarity.

- **Smooth transitions and connections:** Check if the object's transition into the background or other objects is seamless and natural, without awkward or forced connections. For example, the object should appear logically integrated into its environment, with smooth connections to surrounding elements.

Output format:

Please output your answer in non-segmented text format, not Markdown format, as follows: `"This is a real image. [Explanation of the realistic aspects found in the image, such as texture authenticity, consistent shapes, accurate lighting, etc.]"`

# Fake

<image>Describes unrealistic areas of synthetic object imagery.

Note:

- **Material and texture anomalies:** Look for objects where the surface texture does not match reality, such as a leaf that looks more like clay or a keyboard with distorted key textures. Ensure the textures are clear and realistic, reflecting the true material properties.

- **Irregular shapes and structures:** Check if objects have unnatural or distorted shapes, like a keyboard with a misshapen spacebar or a boat stacked on another boat. Ensure that the structure and geometry of the object align with what is physically plausible.

- **Color and lighting distortions:** Watch for overly saturated colors that make objects appear unrealistic, such as an overly bright or unnatural color palette. Ensure that lighting and shadows are applied in a consistent and believable way.

- **Blurry and unclear details:** Look for details that are fuzzy or difficult to interpret, such as unreadable text on an electronic device or unclear edges on objects. Ensure that the object's features are sharp and well-defined.

- **Unnatural transitions and connections:** Check if the object's transition to the background or other objects seems awkward or unnatural, such as an illogical connection between the ocean and a distant mountain. Ensure all object transitions and connections are smooth and realistic.

Output format:

Please output your answer in non-segmented text format, not Markdown format, as follows: `"This is a fake image. [Explanation of the issues found in the image, such as texture anomalies, irregular shapes, lighting inconsistencies, etc.]"`

# Real

<image>Describes realistic areas of real human portrait imagery.

Note:

- **Texture clarity and definition:** Check if the skin, hair, and other body parts have clear, well-defined textures, with no unnatural blending or merging. Ensure the edges between areas, like fingers and hair, are sharp and distinct, reflecting real-world detail.

- **Natural proportions and symmetry:** Ensure that all body parts are proportioned correctly and symmetrically, following realistic anatomical rules. For example, fingers should not be too long, eyes should be aligned, and the mouth should have natural symmetry.

- **Accurate color and saturation:** Watch for natural skin tones and realistic clothing colors, avoiding overly saturated or unnatural hues. Ensure the transitions between different areas of the image, such as between skin and clothing, are smooth and true to life.

- **Sharp details and no artifacts:** Ensure that all details, such as teeth, pupils, and small facial features, are crisp and clear. There should be no noticeable artifacts, such as blurry features or missing details, like a poorly rendered neck.

- **High image quality:** Check if the image maintains a high level of clarity, with no visible noise, distortion, or blurring in the foreground or background. The image should be uniformly sharp, with consistent quality throughout.

- **Logical placement and accurate perspectives:** Ensure that all elements within the image, including clothing, hair, and body parts, are correctly positioned according to realistic perspectives and lighting. There should be no illogical placement, such as mismatched highlights or shadows on clothing.

Output format:

Please output your answer in non-segmented text format, not Markdown format, as follows: "This is a real image. [Explanation of the realistic aspects found in the image, such as clear textures, accurate proportions, natural color, etc.]"

# Fake

<image>Describes unrealistic areas of synthetic human portrait imagery.

Note:

- **Texture blending and blurring:** Look for blurred edges or unnatural fusion between different parts of the image, such as fingers merging or hair blending into the skin. Ensure that details like skin, hair, and fingers are clearly defined and separated, without textures merging unnaturally.

- **Structural distortion and asymmetry:** Check for unnatural proportions or distorted body parts, such as a finger that is too long, an asymmetrical mouth, or eyes that are misaligned. Ensure all body parts are correctly positioned and proportioned, following realistic anatomical rules. Ensure that the structure and geometry of the object align with what is physically plausible.

- **Color and saturation anomalies:** Watch for overly saturated colors or unrealistic transitions, such as unnatural skin tones or overly vivid clothing details. Ensure the overall color scheme and texture transitions appear natural and true to life. Ensure that lighting and shadows are applied in a consistent and believable way.

- **Detail errors and artifacts:** Look for unnatural details, such as odd textures on the teeth or irregularities in the pupils, and check for missing or incomplete details like poorly rendered necks or blurry features. Ensure all details are sharp and well-defined, with no noticeable artifacts or unrealistic elements.

- **Overall quality issues:** Check if the image quality is compromised, such as noticeable noise, image distortion, or a blurry foreground/background. Ensure the image maintains a high level of clarity, with a consistent level of sharpness across the image.

- **Object placement and logical errors** Look for any errors in the placement or logic of the image, such as clothing with mismatched highlights and shadows or an incorrect position of elements like skirts or hair. Ensure that all elements within the image align logically with realistic perspectives and lighting.

Output format:

Please output your answer in non-segmented text format, not Markdown format, as follows: "This is a fake image. [Explanation of the issues found in the image, such as texture blending, structural errors, color anomalies, etc.]"

# Real

<image>Describes realistic areas of real satellite imagery.

Note:

- **Clear outlines of buildings and vehicles:** Check if the outlines of buildings and vehicles are sharp and well-defined, without blurring or distortion.

- **Organized arrangement of objects:** Check if objects on the ground are arranged in a consistent and orderly fashion, with a logical spatial layout.

- **Natural boundary transitions:** Observe if the boundaries between vegetation, land, and buildings are clear, with smooth and realistic transitions.

- **Realistic shadows and lighting:** Check if shadows are positioned logically, following physical laws, and are consistent with the light sources in the image.

- **Natural textures and curves:** Look for textures and curves that are typical in real satellite images, with no unusual patterns or distortions.

- **Proper alignment of roads:** Check if the roads are aligned and show natural curves, without any distortion, misalignment, or unrealistic interruptions.

Output format:

Please output your answer in non-segmented text format, not Markdown format, as follows: "`This is a real image. [Explanation of the realistic aspects found in the image, such as clear outlines, organized objects, realistic shadows, aligned roads, etc.]`"

# Fake

<image>Describes unrealistic areas of synthetic satellite imagery.

Note:

- **Blurry outlines of buildings and vehicles:** Check if the outlines of buildings and vehicles are clear, especially if their edges are blurred or distorted.

- **Disorganized arrangement of objects:** Check if objects on the ground are arranged irregularly, appearing chaotic and lacking logical spatial layout.

- **Unnatural boundary transitions:** Observe if the boundaries between vegetation, land, and buildings are blurred, with any texture mixing or unnatural transitions.

- **Unrealistic shadows and lighting:** Check if shadows appear in illogical positions, especially whether shadows of trees or buildings follow physical laws.

- **Abnormal textures and curves:** CLook for any unnatural curves, stripes, or textures that do not match those found in real satellite images.

- **Distortion or misalignment of roads:** Check if the roads in the image are distorted or misaligned, particularly if road lines are clear, the direction appears natural, or if there are unreasonable curves or interruptions.

Output format:

Please output your answer in non-segmented text format, not Markdown format, as follows: "`This is a fake image. [Explanation of the issues found in the image, such as blurry outlines, disorganized objects, unrealistic shadows, misaligned roads, etc.]`"

# Real

<image>Describes realistic features of real document images.

Note:

- **Coherent and legible text:** Check if the text in the document is semantically coherent, logically structured, and meets standard readability, avoiding nonsensical or illogical phrases.
- **Well-aligned text layout:** Check if the text is arranged in a neat and organized manner, mimicking the alignment and structure of genuine documents with proper margins, spacing, and indentation.
- **Natural fonts and typography:** Observe if the fonts appear consistent and realistic, without any noticeable distortions, deformations, or irregularities.
- **Consistent formatting and structure:** Look for logical paragraph breaks, appropriate headers, and overall formatting that reflects a professionally produced document.
- **Attention to typographic details:** Verify the clarity of characters, letter spacing, and overall text sharpness, ensuring the document appears authentic and naturally printed.

Output format:

Please output your answer in non-segmented text format, not Markdown format, as follows: `"This is a real image. [Explanation of the realistic aspects found in the document, such as coherent text, well-aligned layout, natural fonts, and consistent formatting.]"`

# Fake

<image>Describes unrealistic features of fake document images.

Note:

- **Incoherent and illegible text:** Check if the text in the document lacks semantic coherence, contains nonsensical or illogical phrases, and fails to meet standard readability, making it appear artificial.
- **Misaligned and disorganized text layout:** Check if the text arrangement appears cluttered, lacks proper margins, spacing, or indentation, and does not follow the structured alignment of genuine documents.
- **Unnatural fonts and typography:** Observe if the fonts exhibit inconsistencies, distortions, deformations, or irregularities, making the document look artificial or computer-generated.
- **Inconsistent formatting and structure:** Look for unnatural paragraph breaks, misplaced headers, or formatting anomalies that disrupt the logical flow of a professionally produced document.
- **Lack of typographic detail:** Verify if the characters appear unclear, have irregular letter spacing, or exhibit a lack of sharpness, reducing the overall authenticity of the document.

Output format:

Please output your answer in non-segmented text format, not Markdown format, as follows: `"This is a fake image. [Explanation of the unrealistic aspects found in the document, such as incoherent text, misaligned layout, unnatural fonts, and inconsistent formatting.]"`

## face manipulation

# Real

<image> You are verifying a KNOWN AUTHENTIC photo. Confirm:

- **Nostril Geometry:** Natural asymmetric shape with pore-level detail.
- **Skin Texture:** Gradual tone transitions with microscopic skin imperfections.
- **Eye Reflection:** Physically accurate light interactions in cornea and conjunctiva.
- **Lip Texture:** Visible lip striations with natural moisture gradient.
- **Shadow Integrity:** Consistent ambient occlusion in nasal folds and facial contours.
- **Biological Signatures:** Micro-movements in facial muscles and natural blink patterns.

Output format:

Please output your answer in non-segmented text format, not Markdown format, as follows: `"This is a real image. [Authenticity evidence]"`

Examples:

Please give me your analysis of the given picture.

ASSISTANT:

# Fake

<image> You are analyzing a KNOWN SYNTHETIC image. Check these artifacts:

- **Nostril Area:** Identify blurriness, pixelation, or AI-generated irregularities.
- **Skin Texture:** Detect artificial color transitions, wax-like smoothing, or patchy texture.
- **Eye Region:** Find unnatural reflections, mismatched pupil details, or AI-generated artifacts.
- **Lip Contours:** Check for bleeding colors, mismatched texture, or edge inconsistencies.
- **Facial Contours:** Identify over-smoothed jawlines, unnatural shadow transitions, or warping.
- **Global Consistency:** Detect mismatched lighting direction, floating hairs, or texture repetition.

Output format:

Please output your answer in non-segmented text format, not Markdown format, as follows: `"This is a fake image. [Specific technical artifacts]"`

Please give me your analysis of the given picture.

ASSISTANT:

