# OpenReview forum: "Spot the Fake: Large Multimodal Model-Based Synthetic Image Detection with Artifact Explanation"
_NeurIPS.cc/2025/Conference — NeurIPS 2025 poster_

### Official Review · Reviewer_uBJy · 2025-06-29

**Clarity:** 4
**Significance:** 3
**Originality:** 3
**Rating:** 6
**Confidence:** 5

**Summary:**

This paper addresses the increasingly pressing challenge of detecting AI-generated images by introducing FakeVLM, a large multimodal model (LMM) that not only distinguishes between real and synthetic images with high accuracy but also explains the detected artifacts in natural language, thereby markedly improving interpretability. To train and evaluate the model, the authors construct FakeClue, a large-scale, multi-category dataset of more than 100 000 images. FakeClue is annotated via an innovative multi-LMM pipeline that generates fine-grained textual descriptions of synthetic artifacts. Experiments show that FakeVLM surpasses general-purpose LMMs and rivals task-specific expert models across multiple benchmarks, establishing a strong and interpretable baseline for synthetic-image forensics.

**Questions:**

See weaknesses.

**Ethical Concerns:**

["NO or VERY MINOR ethics concerns only"]

**Final Justification:**

The authors provided detailed responses that thoroughly addressed all my concerns. I am therefore raising my rating to 6.

**Limitations:**

Yes

**Quality:**

4

**Strengths And Weaknesses:**

Strengths

1. The paper tackles the important problem of enhancing the interpretability of synthetic-image detection and articulates a clear research motivation and significance.

2. The proposed method is simple yet extensible: it employs LMMs to construct an automatic annotation pipeline, enabling the creation of a large, diverse dataset.

3. Extensive experiments demonstrate that the approach is competitive with both expert models and general-purpose LMMs, and a broad set of comparisons is provided.

Weaknesses

1. The current presentation of the FakeClue dataset lacks comprehensive visual exemplars that cover the entire spectrum of image types.

2. Most experiments appear to be conducted on relatively idealized datasets, raising concerns about FakeVLM’s robustness under image degradation or other real-world distortions.

3. FakeClue spans seven categories (animals, humans, scenes, satellites, documents, etc.). It is unclear whether FakeVLM performs uniformly well across all categories; category-specific biases might compromise the model’s general applicability.

---

> ### Author Rebuttal · Authors · 2025-07-31
>
> **W1:** Thank you for your valuable suggestion regarding the inclusion of more visual examples. We completely agree that comprehensive visualizations are crucial for demonstrating the diversity and annotation quality of our FakeClue dataset. In the current submission, we have provided some initial visual cases in **Figure 4** and **Figure 5**. According to this year's rebuttal policy, we are unable to provide additional images or links during the rebuttal stage. Therefore, in the final version of the paper, we will add more comprehensive visual cases to the appendix to showcase representative images and their corresponding artifact annotations for all seven categories in the FakeClue dataset.
>
> **W2:** This is a very important question, as a model's robustness is key to its practical value in real-world scenarios. We have already conducted a series of robustness studies in **Appendix B**, testing the model's performance against common distortions such as JPEG compression, image scaling, and Gaussian noise. The results show that even without specialized training on such distorted data, FakeVLM maintains high performance, demonstrating its resilience to common distortions.
>
> Furthermore, in our response to **Reviewer aBzy**'s **W1** comment, we tested the impact of even more perturbations on our model. The results from that expanded study also show that FakeVLM exhibits strong resilience to these distortions, further highlighting its robustness and practical value.
>
> **W3:** Thank you for your comment. We are also very concerned about whether the model performs uniformly across different categories. In **Appendix C**, we have provided a detailed category-wise performance  of FakeVLM and other models on the LOKI dataset. The results in **Table 8** show that our model, FakeVLM, achieves performance far superior to other models in almost all categories (e.g., scene, animal, object, document, satellite) and exhibits relatively balanced performance. In contrast, models like GPT-4o show a more pronounced category bias (e.g., performing well on the 'Human' category but poorly on 'Satellite'). This set of data provides strong evidence that, thanks to its training on the diverse FakeClue dataset, our model possesses stronger cross-category generalization capabilities and that the issue of bias has been well mitigated.
>
> Furthermore, to more comprehensively address your concern, we  provide an additional performance evaluation of FakeVLM on our FakeClue test set, broken down by its seven categories, as shown below:
>
> ||Human|Satellite|Doc|Deepfake|Scene|Object|Animal|Overall Acc|
> |-|-|-|-|-|-|-|-|-|
> |FakeVLM|99.5|100|99.8|96.1|98.9|98.9|99.1|98.6|

---

> > ### Comment · Reviewer_uBJy · 2025-08-07
> >
> > The authors provided detailed responses that thoroughly addressed all my concerns. I am therefore raising my rating to 6.

---

> > > ### Author Response · Authors · 2025-08-07
> > >
> > > Dear Reviewer uBJy,
> > >
> > > Thank you for your positive feedback and for raising the rating. We are glad our responses addressed your concerns. Your valuable suggestions have been very helpful for strengthening our work.
> > >
> > > As promised, we will incorporate the discussed changes into the final version.
> > >
> > > Thank you again for your time and constructive review.
> > >
> > > Sincerely,
> > >
> > > The Authors

---

### Official Review · Reviewer_vBvp · 2025-07-01

**Clarity:** 3
**Significance:** 3
**Originality:** 2
**Rating:** 4
**Confidence:** 5

**Summary:**

This paper introduces the FakeClue dataset, comprising 100K images with fine-grained artifact annotations, designed to facilitate explanation of deepfake detection using multimodal large language models (MLLMs). Based on this dataset, the authors fine-tune LLaVA-1.5-7B and conduct a series of experiments to demonstrate its effectiveness.

**Questions:**

Please see weakness

**Ethical Concerns:**

["NO or VERY MINOR ethics concerns only"]

**Final Justification:**

The authors' response has adequately addressed my concerns; therefore, I have raised my score to Borderline Accept.

**Limitations:**

Please see weakness

**Quality:**

2

**Strengths And Weaknesses:**

Strengths
1. The construction pipeline of FakeClue is well-defined and task-oriented. Integrating category-specific knowledge into prompt engineering is a good insight. And Section 3 would benefit from including additional category-specific annotation examples.
2. The exploration of MLLMs for deepfake detection is insightful. Section 4.1 shows that using frozen MLLM features with a classification head outperforms a simple Real/Fake question-answering setting. This highlights the inherent difficulty of aligning complex visual content with binary responses, and supports the rationale for leveraging MLLMs for explanation-based detection.

Weaknesses
1. The quality of annotations generated by open-source MLLMs is questionable. Prior works such as FakeBench have demonstrated that open-source models lag significantly behind commercial LMMs (e.g., GPT-4o) in interpretability. The paper should include further visualization or discussion to support that the aggregated annotations from open-source MLLMs can achieve comparable quality to those from stronger proprietary models.
2. The methodological novelty of the paper is limited. Section 4 largely reiterates the architecture of existing LLaVA models. While explaining deepfake detection via MLLMs is a promising direction, simply fine-tuning existing architectures on a new dataset is insufficient. The paper would be significantly strengthened by introducing task-specific modules or architectural innovations.
3. The experimental section requires improvement. First, the set of expert baselines in Table 2 is outdated and limited. Important recent methods such as AIDE (ICLR 2025), HEIE (CVPR 2025), and C2P (AAAI 2024) should be included for a fair comparison. Second, the inclusion of FWA and CDFA in Table 5 seems irrelevant to deepfake detection—was this a mistake? Lastly, to justify the value of explanation-based tasks, the paper should compare simple Real/Fake QA against explanatory QA.
4. The category design in FakeClue is problematic. It is unclear why “human” and “deepfake” are treated as separate categories. As an academic term, “deepfake” should not serve as a categorical label. Moreover, the appendix reveals that prompts provided for “human”, with no corresponding prompts for “deepfakes”.

---

> ### Author Rebuttal · Authors · 2025-07-31
>
> **W1:** Thank you for your question. As you noted, we acknowledge that individual open-source LMMs may have a gap in explanatory capability compared to top-tier closed-source models. However, using closed-source models to annotate such a large dataset would significantly increase costs and reduce the scalability of our entire framework.
>
> Recognizing this issue, we did not rely on any single model. Instead, we developed a carefully designed multi-LMM annotation pipeline (as shown in **Figure 1**), which incorporates several measures to enhance performance. These include: designing prompts with information on common artifacts for different image categories; providing customized prompts based on the image's category and authenticity label; and, after obtaining annotations from multiple models, using a powerful model with structured instructions to extract consensus and filter noise. Through these measures, we significantly unlocked the models' potential, improving the stability and accuracy of the annotations.
>
> To directly and quantitatively demonstrate the effectiveness of our data synthesis pipeline, we conducted an additional comparative experiment on the annotation quality of different methods. As shown in the table below, we directly compared the textual similarity scores of annotations generated by different methods  on the LOKI benchmark, of which the textual explanations are annotated by humans. We also included GPT-4V used in Fakebench and the latest versions of GPT-4o (2024-11-20) for a comprehensive comparison.
>
> | |Qwen2_vl|internvl2|Deepseek_vl2|GPT-4o (2024-08-06)|GPT-4o(2024-11-20)|GPT-4V|Qwen2_vl*|Internvl2*|Deepseek_vl2*|Aggregated_answer|
> |-|-|-|-|-|-|-|-|-|-|-|
> |ROUGE_L|0.173|0.184|0.169|0.147|0.160|0.135|0.218|0.200|0.192|0.232|
> |CSS|0.432|0.473|0.388|0.354|0.484|0.454|0.612|0.617|0.578|0.632|
>
> \* indicates the use of our optimized prompts customized for different categories and authenticity labels.
>
> As can be seen, while a base open-source model does lag behind top-tier closed-source models, applying our customized prompts allows the open-source model's annotation quality to surpass that of the best-performing closed-source model, GPT-4o (2024-11-20). Furthermore, the quality of annotations after aggregating the outputs from three models shows even further improvement. This strongly demonstrates the effectiveness of our design and the high quality of our annotations.
>
> **W2:** Thank you for your valuable feedback on the methodological novelty. We appreciate the opportunity to more clearly clarify the contributions of our work.
>
> Using MLLMs for interpretable synthetic image detection is an important and relatively new research direction. While some prior works ([1][2][3]) have utilized MLLMs for synthetic image detection, they have focused primarily on the deepfake domain, leaving the task of general synthetic detection underexplored.
>
> Therefore, as described in **Section 4.1**, we first analyzed the limitations of directly applying existing MLLMs to general synthetic detection tasks. Through experiments, we discovered that although the visual features of MLLMs already contain information to distinguish between real and fake, their performance is suboptimal when simply tasked with answering a binary "real/fake" question. This is likely because complex visual artifacts are difficult to align with simple binary text labels. Based on this insight, we proposed and constructed the Explanatory QA training paradigm. We demonstrated through rigorous ablation studies (as shown in **Figure 3** and **Table 6**) that compelling the model to learn 'why an image is synthetic' rather than just 'if an image is synthetic' unlocks its intrinsic fine-grained visual discrimination capabilities, yielding superior performance compared to simple fine-tuning or binary classification. Therefore, although our model is primarily based on the existing LLaVA architecture, through a  combination of our high-quality data synthesis pipeline and an end-to-end Explanatory QA training paradigm, we have achieved performance that surpasses top-tier closed-source LMMs as well as SOTA expert models.
>
> Furthermore, beyond the FakeVLM model, another significant contribution of our work is the construction of the FakeClue dataset. It contains over 100,000 images across seven distinct categories, including animals, humans, scenes, satellites, and documents. More importantly, our data synthesis pipeline enables the low-cost construction and expansion of high-quality datasets, which is crucial for LMMs that inherently benefit from scaling.
>
> We thank you again for your valuable suggestion. We agree that exploring architectural innovations, such as task-specific modules, is an effective path to further strengthen our work and enhance model performance, and we plan to conduct further explorations of the model architecture in our future work.
>
> **W3:**
>
> 1. Thank you for your valuable suggestions. We have carefully reviewed the methods you mentioned, AIDE (ICLR 2025) [4], C2P (AAAI 2024) and HEIE (CVPR 2025). However, we found that the latter two methods have not publicly released their official code or model weights. Consequently, to address your concern, we have added AIDE (ICLR 2025) [4], which you mentioned, as well as two other recent and open-source SOTA models for comparison:  FreqNet (AAAI 2024) [5] and Fatformer (CVPR 2024) [6]. We have conducted a detailed performance comparison between our proposed FakeVLM and these latest methods on the FakeClue and LOKI datasets. The results are as follows:
> |Model|Fakeclue Acc|Fakeclue F1|LOKI Acc|LOKI F1|
> |-|-|-|-|-|
> |FreqNet|48.7|39.3|58.9|50.6|
> |Fatformer|54.5|45.1|58.8|48.4|
> |AIDE|59.6|86.2|71.1|82.3|
> |AIDE*|85.9|94.5|65.6|80.2|
> |FakeVLM|98.6|98.1|84.3|83.7|
>
>      \* represents the test results of the best checkpoint trained on the FakeClue dataset according to the official default settings; otherwise, the results are from using the officially provided weights.
>
>     As the results show, our model demonstrates a significant performance advantage on both the FakeClue test set and the LOKI benchmark.
>
> 2. Thank you for keenly pointing out the potential issue with the FWA and CDFA comparison methods in Table 5. We sincerely apologize for the confusion, which was entirely due to a citation error on our part. The original references [63] and [66] were incorrect. The correct citations for these methods should be:
>
>     - FWA: Li, Y., & Lyu, S. (2018). Exposing deepfake videos by detecting face warping artifacts.
>     - CDFA: Lin, Y., et al. (2024). Fake it till you make it: Curricular dynamic forgery augmentations towards general deepfake detection.
>
>     FWA is a classic method for detecting face warping artifacts, while CDFA is a recent SOTA method in general deepfake detection. We once again offer our sincere apologies for any misunderstanding caused by our citation oversight and will correct the relevant citations and descriptions in the final version to ensure the rigor and clarity of our experimental comparisons.
>
> 3. We strongly agree that these two comparisons are of great importance and have already conducted this key ablation study in our paper. In **Section 4.1** and **Figure 3**, we explicitly compare four different methods, including the Real/Fake QA and Explanatory QA you mentioned. Furthermore, additional results from this four-method comparison (on both FakeClue and LOKI) can be found in **Appendix A**.
>
> **W4:** The primary reason we separated "Human" and "DeepFake" in our paper is due to their different data sources and distinct types of common artifacts. The "DeepFake" category in our dataset specifically refers to face forgery images from the FF++ dataset. These are primarily created by manipulating and replacing the face in a real image, so the artifacts are mainly concentrated on facial features (eyes, nose, mouth, etc.). In contrast, the "Human" category is sourced from datasets like GenImage, which contain fully synthetic images. These images encompass a wider variety of synthetic content featuring human subjects, and their artifact patterns may include distorted body structures or unnatural skin, which differ from classic deepfakes.
>
> We agree that using "deepfake" as a categorical label is not sufficiently rigorous from an academic standpoint. Therefore, we will replace it with "Face Manipulation" in the final version. We will also add the missing prompt for this category to the appendix.  This prompt will guide the model to assess authenticity by analyzing key indicators. For fake images, the model will focus on detecting: (1) blurry or inconsistent face boundaries; (2) mismatches in lighting, color, or texture between the face and body; and (3) distortions in key features such as the eyes or teeth. For real images, it will emphasize: (1) smooth transitions between the face, hairline, and neck; (2) coherent lighting and shadows; and (3) clearly defined facial details.
>
> *References:*
>
> [1] Zhang, Y., Colman, B., Guo, X., Shahriyari, A., & Bharaj, G. (2024, September). Common sense reasoning for deepfake detection. In ECCV (pp. 399–415). Cham: Springer.
>
> [2] Huang, Zhengchao, et al. "Ffaa: Multimodal large language model based explainable open-world face forgery analysis assistant." arXiv preprint arXiv:2408.10072 (2024).
>
> [3] Chen, Yize, et al. "X2-dfd: A framework for explainable and extendable deepfake detection." arXiv preprint arXiv:2410.06126 (2024).
>
> [4] Yan, Shilin, et al. A Sanity Check for AI-generated Image Detection. In ICLR, 2025.
>
> [5] Tan, C., Zhao, Y., Wei, S., Gu, G., Liu, P., & Wei, Y. (2024, March). Frequency-aware deepfake detection: Improving generalizability through frequency space domain learning. In AAAI (Vol. 38, No. 5, pp. 5052–5060).
>
> [6] Liu, H., Tan, Z., Tan, C., Wei, Y., Wang, J., & Zhao, Y. (2024). Forgery-aware adaptive transformer for generalizable synthetic image detection. In CVPR (pp. 10770–10780).

---

> > ### Comment · Reviewer_vBvp · 2025-08-07
> >
> > The authors' response has adequately addressed my concerns; therefore, I have raised my score to Borderline Accept.

---

> > > ### Author Response · Authors · 2025-08-07
> > >
> > > Thank you for re-evaluating our work and raising our score. We are very glad that our response successfully addressed your concerns.
> > >
> > > We sincerely appreciate your time and valuable feedback.

---

### Official Review · Reviewer_aBzy · 2025-07-02

**Clarity:** 3
**Significance:** 3
**Originality:** 3
**Rating:** 5
**Confidence:** 4

**Summary:**

This paper proposes FakeVLM, a fully fine-tuned large multimodal model based on LLaVA-1.5-7B. By pairing a CLIP-ViT visual encoder with a Vicuna language head, the model judges the authenticity of images and provides natural-language explanations of visual artifacts. The authors also construct the FakeClue dataset, which covers seven categories—animals, humans, scenes, satellite images, and others—and contains over 100,000 real and synthetic images. Fine-grained artifact descriptions are generated through multi-model collaborative annotation and rule-based voting. Experiments show that FakeVLM outperforms strong baselines such as GPT-4o and Qwen2-VL on benchmarks including FakeClue, LOKI, DD-VQA, and DMImage, and remains robust under common distortions such as JPEG compression, resizing, and Gaussian noise. The released model weights and dataset provide a unified and interpretable framework for synthetic-image detection.

**Questions:**

1. What are the main differences in the annotation prompts for different categories, and how were they designed?
2. How are the outputs of the three models aggregated? What happens when their predictions differ?
3. Clarifying whether different categories focus on different types of artifacts and what the main differences are would also help understanding.

**Ethical Concerns:**

["NO or VERY MINOR ethics concerns only"]

**Final Justification:**

My concerns and questions have been addressed.

**Limitations:**

Yes

**Paper Formatting Concerns:**

Not found.

**Quality:**

3

**Strengths And Weaknesses:**

**Strengths:**
1. This work addresses an important interpretability issue in synthetic-image detection with a simple yet effective approach, enhancing explainability.
2. The authors release FakeClue, a new dataset with more than 100,000 annotated samples from diverse domains.
3. The method is compared in detail across multiple datasets and models.
4. Experimental results show that the approach is highly competitive, outperforming large general-purpose LMMs and several domain-specific expert models.

**Weaknesses:**
1. Only mild JPEG compression, resizing, and Gaussian noise are tested; additional common perturbations should be included to fully assess model stability.
2. The process of deriving final annotations from the outputs of three models is insufficiently detailed; more specifics are needed.
3. The paper lacks a description of how the category-specific annotation prompts were obtained.

---

> ### Author Rebuttal · Authors · 2025-07-30
>
> We sincerely thank you for your comprehensive review and positive assessment of our work. Your specific questions are very helpful for us to further clarify the details of our work.
>
> **W1:** This is a very valuable suggestion. Our robustness study in **Appendix B (Table 7)** was intended as a preliminary reference. To conduct a more comprehensive evaluation as you suggested, we have expanded this study to include more diverse and rigorous perturbations. The results for these additional perturbations are as follows:
> | |ACC|F1|ROUGE_L|CSS|
> |-|-|-|-|-|
> |flip_horizontal|98.42|98.28|54.04|83.28|
> |rotate15|90.88|89.58|43.95|76.55|
> |sharpen1.5|97.36|97.18|54.26|83.55|
> |contrast0.7|96.44|96.07|52.96|82.79|
> |contrast1.3|98.18|98.03|53.98|83.32|
> |blur3|89.50|87.76|50.32|81.15|
> |origin image|98.60|98.10|58.00|87.70|
>
> The new perturbations include common image distortions such as rotation, flipping, sharpening, contrast adjustment, and Gaussian blur. Despite not being specifically trained on distorted data, FakeVLM demonstrates resilience to these distortions, which highlights the model's robustness and practical value.
>
> **W3&Q1&Q3:** The core difference in the annotation prompts we designed for various categories lies in the distinct points of focus and common artifact types that we predefined for each category. This design is based on prior knowledge of common generation errors for different image types, which we derived from a summary of existing literature [1] and our own empirical observations of various synthetic images.
>
> For instance, for the Human category, the prompt guides the model to focus on details such as "skin color transitions," "eyes and eyebrows area," "lips and mouth," and "overall facial contours." In contrast, for the Satellite category, the prompt directs the model to focus on "the clarity of building and vehicle outlines," "the alignment of roads," and "the realism of shadows."
>
> Based on this approach, we designed a total of 12 distinct prompt templates for these categories and the two authenticity labels (real/fake). We have provided detailed prompts for all categories in **Appendix G** of our paper.
>
> **W2&Q2:**
>
> **The details of our aggregation process:**  As described in **Section 3.2**, we first use three different high-performing LMMs to generate three candidate artifact descriptions for each image. Then, we employ a fourth model, Qwen2-VL, as an aggregator. This aggregator model is given the three candidate descriptions along with an instruction prompt containing detailed guidelines (see **Appendix G** for details) to produce the final, unified annotation.
>
> **Handling prediction differences:** In our pipeline, we first obtain the ground-truth authenticity label from the original dataset. As mentioned in **Section 3.2 (lines 136-138)**, these pre-processed labels serve as prior knowledge to guide the annotation. Therefore, the authenticity (real/fake) label of a known image is not subject to misjudgment by design. For cases where the artifact descriptions differ, our aggregation prompt provides explicit guidance. It instructs the aggregator model to extract common points and filter out minority opinions (unless the artifact is particularly obvious and important). In this way, we ensure that the final annotation is the result of a consensus among multiple models, making it more objective and stable.
>
> *Reference:*
>
> [1] Mathys, M., Willi, M., & Meier, R. (2024). Synthetic photography detection: A visual guidance for identifying synthetic images created by ai. arXiv preprint arXiv:2408.06398.

---

### Official Review · Reviewer_q4Ca · 2025-07-02

**Clarity:** 4
**Significance:** 3
**Originality:** 3
**Rating:** 5
**Confidence:** 3

**Summary:**

This study is focused on creating a diverse dataset, namely FakeClue, and use the new dataset to train a new generalized detector both synthetic and deepfake images. The new dataset is derived from a number of published datasets, like FF++, GenImage, CVUSE, ViGOR, etc with additional category labels in 7 categories, and captions that are generated by fusing Qwen2-VL, InterVL, DeepSeek captions that explain the justification of real/fake labels.  While it is generally risky to use ground truth that is strictly based on ML models which were not trained on the specific task, Section 4.1 is a good ablation study to establish that the FakeClue labels align well with human labels after the explanations.

The diversity of the dataset is particularly valuable to avoid overfitting to smaller datasets for simpler tasks, like only detecting fake faces. The model, which is based on the existing LLAVA 1.5, is trained on FakeClue seems to match or outperform the benchmarks models while being tested on the other public datasets. It is intuitive to expect that with more data, and more instructive labels with categories and explanations, the trained model performs better.

Overall, this is a solid dataset paper;  the study systematically constructs a diverse datasets with categories, fake/real labels, and human understandable explanations for fake/real labels. The results show that FakeVLM model trained by the new dataset outperforms the existing SOTA models for synthetic image detection. The DeepFake detection results on FF++ also look impressive.

**Questions:**

1. The FakeClue dataset generates its labels combining the three major pre-trained models, Qwen2-VL, InterVL, and DeepSeek. If there are synthetic images in the dataset that are not detectable by all three, the dataset will contain incorrect labels. A detector trained by this dataset will learn to label such synthetic images as real and have a possibly bigger gap in detecting such images. How would the authors think about generalizing beyond the capabilities of these major pre-trained models?

2. An adversarial attacker may use the same three models to figure out the blind spots for all models that are trained with FakeClue. How would authors think about closing such gaps?

3. The DeepFake detection results for all models are a bit dubious because most models are performing over 95% on FF++ dataset; this indicates a need for a more challenging benchmark for DeepFake detection. Evaluations with CDFv2 or DFDCP may reveal true performance differences, if there are any.

CDFv2: Li , Y., Yang , X., Sun , P., Qi 370 , H., & Lyu , S. (CVPR 2020) Celeb-df: A new dataset for deepfake forensics

DFDCP: Dolhansky , B., Bitton , J., Pflaum , B., Lu , J., Howes , R., Wang , M., & Ferrer , C. C. (Arxiv 2020) The deepfake detection challenge dataset

**Ethical Concerns:**

["NO or VERY MINOR ethics concerns only"]

**Final Justification:**

The authors have answered my questions and further improved the quality of the study. I keep my accept score for this work.

**Limitations:**

Detecting fake images have very important applications in real life, as explained in the paper introduction. The paper contributes a valuable dataset that can be used to improve all synthetic and deep fake detection methods.

**Paper Formatting Concerns:**

No.

**Quality:**

3

**Strengths And Weaknesses:**

Quality:
The work follows a systematic approach to generate a large and diverse dataset, namely FakeClue. The new dataset is used to train a new model, FakeVLM. The ablation studies are clear and well motivated. The results show significant improvements over the existing benchmarks. Overall, this is a solid dataset paper.

Clarity:
The paper is well written and easy to follow.  The results with the existing datasets and the new dataset allow reproduction of results. One typo in 278, DMimage dataset name seems misspelled.

Significance:
I believe the new dataset allows all fake/real classification models to improve; this is a significant dataset contribution.

Originality:
This is a dataset paper; the FakeClue dataset combining a variety of artifacts, categories, multi-LMM labels of fake/real and explanations is a new dataset as explained in Table 1.

---

> ### Author Rebuttal · Authors · 2025-07-30
>
> We sincerely thank you for your positive evaluation and insightful questions. We are glad that you recognized the methodology of our dataset construction, the effectiveness of our FakeVLM model, and the contributions of this work. Your questions are highly valuable to us. Additionally, we thank you for your meticulous review and for pointing out the typo of "DMimage" in line 278. We will correct it in the final version.
> Below, we address your questions one by one:
>
> **Q1:** Thank you for this crucial question regarding label quality and model generalization. We fully agree that the quality of the dataset's labels is fundamental to the model's performance.
>
> First, we would like to clarify that in our FakeClue dataset annotation pipeline, for any given image to be annotated, we first obtain its ground-truth authenticity (Real/Fake) label from the original dataset. Then, as mentioned in **Section 3.2 (lines 136-138)** of our paper, these pre-processed authenticity labels are used as prior knowledge to guide the annotation process. Specifically, for a known fake image, we use specific prompts to guide the LMM to focus on finding and describing artifacts; for a known real image, we guide the LMM to analyze its plausible and realistic features. Therefore, our data annotation pipeline, by design, ensures that known synthetic images are not incorrectly labeled as "real."
>
> To address the other aspect of your question, which concerns potential simultaneous flaws in the explanatory texts from all three models and how our trained model can surpass the pretrained capabilities of the three annotator models, our design incorporates several key points to ensure the quality and reliability of the explanations:
> - Model Diversity: We selected three top-tier open-source LMMs, each with a distinct architecture and training data. This diversity significantly reduces the probability of all three models making the same error on the same artifact in the same image.
> - Injecting Prior Knowledge: Before annotation, we provide the models with prior knowledge, including the authenticity label, image category, and common artifact types for that category. This greatly constrains the model's output and reduces the likelihood of hallucinations.
> - Reliable Aggregation Rule: Our aggregation rule explicitly requires "prioritizing explanations mentioned by at least two models" and "discarding points mentioned by only a single model (unless they are critical artifacts)." This mechanism effectively filters out random errors from individual models, ensuring the consensus and reliability of the final annotations.
>
> Furthermore, as you may refer to our response to **Reviewer vBvp**'s comment **W1**, we have experimentally demonstrated the effectiveness of the above design. Moreover, our experimental results (see **Table 2**) clearly show that the model trained on such data surpasses the original three models on both FakeClue and LOKI datasets . Finally, we completely agree on the importance of continuously improving dataset quality. In the future, we can build upon these measures by introducing manual spot-checks to further enhance the reliability of our dataset.
>
> **Q2:** Thank you again for this highly forward-thinking question. Overall, we believe that finding a universal blind spot for our model would be quite challenging. A key advantage of our FakeClue dataset is its unprecedented diversity. It covers 100,000 data points across 7 distinct categories, sourced from multiple open datasets and our own in-house data for specialized types. This diversity ensures that our model, FakeVLM, learns more general and robust artifact features, rather than flaws specific to a particular generative model or domain. Consequently, it is significantly more difficult for an attacker to find a universal blind spot that can deceive the model across all categories compared to models trained on smaller, single-task datasets.
>
> However, we also acknowledge that such blind spots could potentially exist. As you rightly pointed out, we believe no static dataset or model can  permanently solve the forgery detection problem. Our contribution is not just a model, but an extensible and dynamic framework. When new blind spots or generative models emerge, our data construction pipeline can be utilized to efficiently and cost-effectively integrate new content and artifact patterns into the FakeClue dataset. Subsequently, FakeVLM can be incrementally trained or retrained to continuously close these potential gaps.
>
> **Q3:** We completely agree with your assessment. The FF++ dataset indeed shows a certain saturation trend, with many models achieving very high performance. Our decision to evaluate on FF++ and the FF++-based DD-VQA was primarily because they are widely used and recognized benchmarks in the field of face forgery.
>
> To more comprehensively test the upper limits of FakeVLM's face detection capabilities against more subtle and realistic forgeries, we have evaluated its AUC on CDFv2 and DFDC (we note that DFDCP, as per your reference, is commonly referred to as DFDC). The results are as follows:
> | |FWA [1]|Face X-ray [2]|SRM [3]|RECCE [4]|UCF [5]|FakeVLM|
> |-|-|-|-|-|-|-|
> |CDFv2|66.80|67.86|75.52|68.71|75.27|79.00|
> |DFDC|61.32|76.55|69.95|69.06|71.91|74.45|
>
> The results show that our model achieves the best performance on CDFv2. Furthermore, on the highly challenging DFDC benchmark, which is known for its realistic settings and complex augmentations, our model also achieved a competitive AUC, demonstrating its strong forgery detection capabilities.
>
> *References:*
>
> [1] Li, Yuezun, and Siwei Lyu. "Exposing deepfake videos by detecting face warping artifacts." arXiv preprint arXiv:1811.00656 (2018).
>
> [2] Li, Lingzhi, et al. "Face X-ray for More General Face Forgery Detection, 2020 IEEE." CVF Conference on Computer Vision and Pattern Recognition (CVPR). 2019.
>
> [3] Lee, HyunJae, Hyo-Eun Kim, and Hyeonseob Nam. "Srm: A style-based recalibration module for convolutional neural networks." Proceedings of the IEEE/CVF International conference on computer vision. 2019.
>
> [4] Cao, Junyi, et al. "End-to-end reconstruction-classification learning for face forgery detection." Proceedings of the IEEE/CVF conference on computer vision and pattern recognition. 2022.
>
> [5] Yan, Zhiyuan, et al. "Ucf: Uncovering common features for generalizable deepfake detection." Proceedings of the IEEE/CVF international conference on computer vision. 2023.

---

### Note · Authors · 2025-08-15

Dear Reviewers, Area Chair, and Senior Area Chair,

Following comprehensive responses and discussions with all reviewers, we are sincerely grateful for the productive discussions and thoughtful feedback from all reviewers, which have been invaluable in strengthening our work.

**Review Highlights:**

We are encouraged that the reviewers recognized and valued our key contributions, including:
* **Novel and Diverse Dataset:** The construction of FakeClue, a large-scale, multi-category, and extensible dataset for synthetic image detection with fine-grained artifact annotations.
* **Strong and Interpretable Model:** The FakeVLM model, which achieves state-of-the-art accuracy while providing clear, natural-language explanations for its judgments.
* **Effective Training Paradigm:** The novel Explanatory QA approach that effectively unlocks the fine-grained visual discrimination capabilities of Large Multimodal Models for this task.

**Main Concerns Addressed:**

During the rebuttal, we implemented substantial clarifications and improvements to address reviewers' main concerns:
* **Expanded Evaluation Results:** We added experiments on more challenging benchmarks (CDFv2, DFDC) and against recent SOTA expert models (e.g., AIDE), further demonstrating FakeVLM's superior performance.
* **Comprehensive Robustness Analysis:** We broadened our robustness tests to include a wider array of common image perturbations, confirming the model's practical resilience.
* **Validated Annotation Pipeline:** We provided new quantitative results to validate the high quality of our multi-LMM annotation process, showing it can surpass even leading proprietary models.
* **Commitment to Revisions:** We will revise the manuscript guided by the reviewers' insightful feedback, incorporating key improvements such as refining category definitions and adding more visual examples.

We are grateful for the time and effort invested by the reviewers and chairs in evaluating our work and facilitating a productive exchange during the discussion period. We believe our work is now significantly strengthened and will be a valuable contribution to the community.

Best regards,

The Authors

---

### Decision · Program_Chairs · 2025-09-17

**Decision:**

Accept (poster)

**Comment:**

This paper received overall positive reviews. Three reviewers (q4Ca, aBzy, uBJy) gave Accept or Strong Accept ratings, highlighting the contributions in constructing the FakeClue dataset, developing the FakeVLM model, improving interpretability, and demonstrating robustness across categories. One reviewer (vBvp) initially raised concerns about methodological novelty and baselines but was satisfied after the rebuttal and updated the rating to Borderline Accept. While the paper’s architectural novelty is somewhat limited, its systematic dataset construction, solid empirical results, and practical significance make it a strong submission. I recommend acceptance.